# Low-dose mixtures of dietary nutrients ameliorate behavioral deficits in multiple mouse models of autism

Tzyy-Nan Huang[1]☉, Ming-Hui Lin[1,2]☉, Tsan-Ting Hsu[1], Chen-Hsin Yu[1], Yi-Ping Hsueh [1,2]*

1 Institute of Molecular Biology, Academia Sinica, Taipei, Taiwan, Republic of China, 2 Molecular and Cell Biology, Taiwan International Graduate Program, Institute of Molecular Biology, Academia Sinica and Graduate Institute of Life Sciences, National Defense Medical Center, Taipei, Taiwan, Republic of China

☉ These authors contributed equally to this work.
* yph@as.edu.tw, yph@gate.sinica.edu.tw

## Abstract

Autism spectrum disorder (ASD) is a group of heterogeneous, behaviorally defined neurodevelopmental conditions influenced by both genetic and environmental factors. Here, we show that supplementation of multiple low-dose nutrients—an important environmental factor contributing to ASD—can modulate synaptic proteomes, reconfigure neural ensembles, and improve social behaviors in mice. First, we used *Tbr1*[+/−] mice, a well-established model of ASD, to investigate the effect of nutrient cocktails containing zinc, branched-chain amino acids (BCAA), and serine, all of which are known to regulate synapse formation and activity. Supplementation of nutrient cocktails for 7 days altered total proteomes by increasing synapse-related proteins. Our results further reveal that *Tbr1* haploinsufficiency promotes hyperactivation and hyperconnectivity of basolateral amygdala (BLA) neurons, enhancing the activity correlation between individual neurons and their corresponding ensembles. Nutrient supplementation normalized the activity and connectivity of the BLA neurons in *Tbr1*[+/−] mice during social interactions. We further show that although a low dose of individual nutrients did not alter social behaviors, treatment with supplement mixtures containing low-dose individual nutrients improved social behaviors and associative memory of *Tbr1*[+/−] mice, implying a synergistic effect of combining low-dose zinc, BCAA, and serine. Moreover, the supplement cocktails also improved social behaviors in *Nf1*[+/−] and *Cttnbp2*[+/M120I] mice, two additional ASD mouse models. Thus, our findings reveal aberrant neural connectivity in the BLA of *Tbr1*[+/−] mice and indicate that dietary supplementation with zinc, BCAA, and/or serine offers a safe and accessible approach to mitigate neural connectivity and social behaviors across multiple ASD models.

**Data availability statement:** All data supporting the conclusions of this article are provided within the main text, figures, and Supporting information (including supplementary figures and tables, source data [S3 Data], and original images [S1 Raw Images]). The LC–MS/MS proteomic data are available via the ProteomeXchange Consortium with identifier PXD069666. Custom code is available on GitHub (https://github.com/HsuehYiPing/Tbr1Cocktail) and archived in Zenodo (DOI: 10.5281/zenodo.17394195).

**Funding:** This work was supported by grants from Academia Sinica, Taiwan (AS-IA-111-L01 to Y.-P.H.) and the National Science and Technology Council, Taiwan (NSTC 113-2326-B-001-008 and 114-2326-B-001-005 to Y.-P.H.). The funders had no role in study design, data collection and analysis, decision to publish, or preparation of the manuscript.

**Competing interests:** I have read the journal's policy and the authors of this manuscript have the following competing interests: Y.-P.H. is a member of PLOS Biology's Editorial Board. Y.-P.H. and T.-N.H. are the inventors of the Patent No. US2021/0113610 A1. The other authors declare that no competing interests exist.

**Abbreviations:** AAV, adeno-associated virus; ASD, autism spectrum disorder; BCAA, branched-chain amino acids; BLA, basolateral amygdala; GO, gene ontology; GRIN, gradient index; mTOR, mammalian target of rapamycin; NMDAR, N-Methyl-D-aspartic acid receptor; OE, object exploration; PBS, phosphate-buffered saline; PCA, principal component analysis; RSI, reciprocal social interaction; VCP, valosin-containing protein; WCSS, within-cluster sum of squares.

## Introduction

Autism spectrum disorder (ASD) is a group of highly prevalent neurodevelopmental conditions characterized by two core behavioral symptoms: impairments in social behavior and communication, and the presence of restricted, repetitive behaviors and sensory abnormalities [1,2]. ASD arises from a combination of genetic and environmental factors [3,4] that influence neural development—particularly synapse formation and signaling—ultimately leading to impaired neural connectivity [5,6]. The crosstalk between genetic variations and environmental factors and the outcomes for synaptic functions and neural ensembles are critical issues in ASD research, yet many aspects of this topic remain to be investigated.

Nutrition is a significant environmental factor contributing to ASD [7], with dietary bias and gastrointestinal disturbances being common comorbidities [8,9]. Accordingly, dietary interventions have been proposed as a treatment avenue [8–11]. Importantly, recent studies have indicated convergence of the effects of certain nutrients, including zinc, branched-chain amino acids (BCAA) and serine, on synapse formation and activity [10,12–18]. Dietary supplementation with these nutrients may improve synapse formation and maintenance, as well as promote synaptic responses, to ameliorate neuronal functions (Fig 1) [7].

Specifically, zinc is highly concentrated at synaptic vesicles and co-released with glutamate upon synaptic stimulation [19,20]. Increased concentrations of postsynaptic zinc enhance SRC kinase activity and result in enhanced conductivity of N-Methyl-D-aspartic acid receptor (NMDAR) [19]. In addition, zinc induces protein-protein interactions and condensate formation of multidomain scaffold proteins, including SHANKs and CTTNBP2, at the postsynaptic site [12,18,21]. Thus, zinc is a critical synaptic modulator of the presynapse to postsynapse signal and it facilitates synaptic signaling, remodeling, and dendritic spine maintenance (Fig 1). Apart from synaptic molecules, long-term zinc supplementation enhances ribosomal protein expression and increases protein synthesis in neurons, consequently correcting synapse deficits in *Cttnbp2*$^{+/M120I}$ mice [22]. Moreover, dietary zinc supplementation improves behavioral deficits of mouse models with deficiencies in the *Shank*, *Cttnbp2*, and *Tbr1* genes [12–14,18,23–27], reinforcing evidence for the crosstalk between zinc and the functions of ASD-linked genes.

D-serine, a derivative of L-serine released from neurons and astrocytes [28,29], enhances NMDAR conductivity by binding to the receptor's glycine-binding site [30] (Fig 1). L-serine supplementation to increase intrinsic D-serine levels by providing the parent material to neurons and astrocytes was shown previously to improve the NMDAR response and alleviate the symptoms of patients harboring an *NMDAR* mutation [31]. Thus, L-serine supplementation ameliorated the deficiencies caused by synaptic NMDAR impairment.

BCAA serves as the building material for protein synthesis. Protein synthesis is an essential downstream pathway of synaptic stimulation to control synaptic remodeling and synaptic responses via a feedback mechanism (Fig 1) [7]. BCAA supplementation in the brain has been found to improve neuronal function and the autism-linked

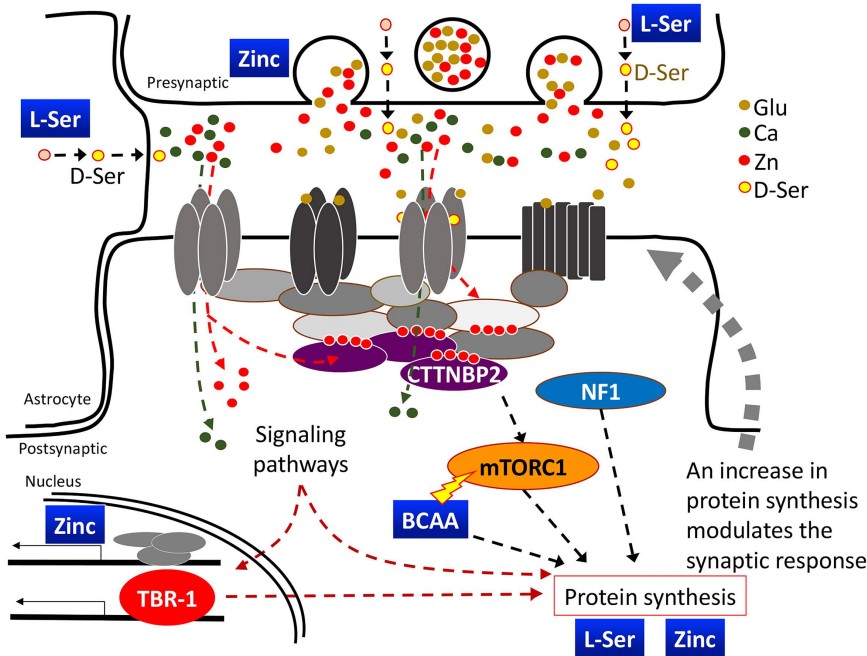

**Fig 1. Zinc, Ser, and BCAA act in concert to regulate synaptic signaling and the synaptic response.** Zinc is highly concentrated at synaptic vesicles and is co-released with glutamate. At the postsynaptic site, zinc induces condensate formation of the postsynaptic proteins and maintains dendritic spine morphology. Synaptic stimulation via the action of glutamate and D-Ser on ion channels activates downstream signaling and promotes protein synthesis, which is required for the long-term synaptic response and synaptic remodeling. BCAA enhances protein synthesis via activation of the mTOR pathway.

phenotypes of mice or patients carrying mutations in *Branched-chain ketoacid dehydrogenase kinase* (*BCKDK*), a gene essential for catabolism of BCAA, and in *Solute carrier transporter 7a5* (*SLC7A5*), a large neutral amino acid transporter essential for maintaining normal levels of BCAA in the brain [11,32,33]. In addition, BCAA also act as a triggering factor to activate the mammalian target of rapamycin (mTOR) pathway for protein synthesis [34–37] (Fig 1). Through this activity, BCAA supplementation ameliorates the impaired dendritic spine formation and synaptic responses caused by various genetic defects, including those displayed by *Nf1*$^{+/-}$, *Cttnbp2*$^{+/M120I}$, and *Vcp*$^{+/R95G}$ mice [14,17,18]. These genes exert distinct molecular functions, yet all are involved in regulating dendritic spine formation or morphology [15,26,38]. Dietary BCAA supplementation increases the density of dendritic spines in mouse brains and improves the social behaviors of several autism mouse models [15–18].

Based on the aforementioned studies, single nutrient supplementation seems a promising and available treatment to improve, though not cure, the ASD-related symptoms in mouse models or patients. Since the diverse array of synaptic processes always interconnect with and influence each other, we hypothesized that apart from individual nutrients, a mixture containing multiple nutrients, including zinc, BCAA, and serine, would prove synergistically beneficial in abrogating ASD-associated symptoms (Fig 1). There are two advantages to using a dietary supplement mixture. First, since these nutrients are all able to regulate synaptic function and morphology, lower doses of individual supplements may be applied to make the mixture, thereby reducing potential side effects associated with high-dose and long-term supplementation of a specific nutrient. The second is the broadened targets of the combined supplements. As long as impaired synaptic function or formation is associated with a target, a supplement mixture containing zinc, BCAA, and serine may ameliorate to a certain extent the ASD-linked deficits. Thus, the applications may not be limited to a specific mutation.

In this report, we investigate the effects of nutrient cocktails containing lower doses of zinc, BCAA, and serine on *Tbr1*[+/−] mice, a well-characterized ASD model [39–42]. We designed three sets of experiments to characterize the effects of our nutrient supplements. First, proteomic analysis was performed to investigate altered protein expression in the brain. Second, in vivo calcium imaging was conducted to dissect the neural ensembles of the basolateral amygdala (BLA), a brain area sensitive to *Tbr1* haploinsufficiency [39]. Previously, we showed that *Tbr1* haploinsufficiency reduces contralateral axonal projection but increases mistargeting of ipsilateral axonal projection of the BLA to other brain regions and modulates functional connectivity at a whole-brain scale [43]. However, neural ensembles at the BLA upon social stimulation have not yet been explored. Thus, using *Tbr1*[+/−] mice as a model, we analyzed the effect of *Tbr1* haploinsufficiency on BLA neural ensembles, and then characterized if our dietary supplements could correct the defective neural ensembles in the BLA of *Tbr1*[+/−] mice. Third, we performed behavioral assays to monitor the beneficial effects of dietary supplementation on the social behaviors of *Tbr1*[+/−] mice. These three sets of analyses uncover the beneficial effects of the supplement cocktails on *Tbr1*[+/−] mice.

Apart from *Tbr1*[+/−] mice, we also analyzed the social behaviors of two additional ASD mouse models, *Nf1*[+/−] and *Cttnbp2*[+M120I] mice. Both *Nf1* deficiency and *Cttnbp2* M120I mutation impair dendritic spine formation through different underlying mechanisms. NF1 protein regulates the function of valosin-containing protein (VCP) in controlling endoplasmic reticulum formation and protein synthesis efficiency, consequently influencing dendritic spine formation [15,16,38]. Leucine supplementation, which increases protein synthesis by activating the mTORC1 pathway, enhances dendritic spine formation and social behaviors in *Nf1* and *Vcp* mutant neurons [15,16]. CTTNBP2 is an intrinsically disordered protein that forms condensates at dendritic spines [13]. It binds cortactin and zinc to regulate cortactin-F-actin dynamics at dendritic spines and zinc homeostasis in the mouse brain [44,45]. Although the ASD-linked M120I mutant of CTTNBP2 exhibits a reduced ability to form condensates at dendritic spines, zinc can still bind this mutant and induce liquid-to-gel phase transition and synaptic retention of the M120I mutant protein [13]. Moreover, zinc supplementation in drinking water improves social behavior deficits of *Cttnbp2*[+/M120I] mice [13,18,22]. Given that synaptic morphology and function and the behavioral deficits of both *Nf1*[+/−] and *Cttnbp2*[+/M120I] mice are improved by individual nutrient supplementation, they represent good candidates for investigating whether our combinatory dietary supplements also exhibit beneficial effects in other mouse models. Together, our study provides evidence for potential therapeutic treatments of ASD-linked deficiencies using dietary supplementation with multiple nutrients.

## Results

### A cocktail of dietary supplements alters total proteomes

The concentrations of single supplements used to improve ASD-linked phenotypes in previous studies are equal to 2% serine, 1.8% leucine, or 40 ppm zinc in drinking water (approximal 4.8 mL per day per mouse) [10,14,15,17]. In the current study, we lowered the concentrations of individual supplements to make our cocktail. Given that L-serine has to be processed into D-serine to modulate neuronal activity, we maintained the L-serine concentration at a relatively higher level than that of BCAA. The cocktail containing 0.45% BCAA (leucine 0.225%, isoleucine 0.1125%, and valine 0.1125%, i.e., Leu:Ile:Val = 2:1:1), 1% serine and 20 ppm zinc is termed "1/4 cocktail", in which the concentration of BCAA was reduced to one-quarter the original dose and the amounts of L-serine and zinc were reduced to half the original doses. The estimated total intake amounts of individual nutrients from drinking water and chow are detailed in Methods.

Our previous studies have indicated that administration of either zinc or BCAA for 1 week alters the synaptic proteomes of *Nf1*[+/−] and *Cttnbp2*-deficient mice [14,16,18]. Although the concentrations of the individual supplements in our supplement cocktails were much lower than provisioned in those previous studies, the cocktail treatments may still modulate brain proteomes to some extent if the supplement cocktails have a synergistic effect on the brain. To investigate that possibility, we analyzed total lysates of whole mouse brains using label-free liquid chromatography–tandem mass spectrometry (LC–MS–MS). Four groups of animals, i.e., WT_water, *Tbr1*[+/−]_water, WT_1/4 cocktail, and *Tbr1*[+/−]_1/4 cocktail, with

three mice in each group were analyzed. We identified ~3,000 protein species from each sample and applied principal component analysis (PCA) to analyze the differences among the four experimental groups (Fig 2B). PC1 did not strongly differentiate among the four groups of data, but the *Tbr1$^{+/-}$*_water group was better separated from the other three groups along PC2 (Fig 2B), indicating that the total proteome of the *Tbr1$^{+/-}$*_water group is relatively different from those of the other three groups. This outcome also indicates that cocktail supplementation renders *Tbr1$^{+/-}$* mice more comparable to WT mice in terms of protein expression profiles of the brain (Fig 2B).

To further characterize the proteomes, we performed network analysis using a Python package for weighted gene co-expression network analysis (PyWGCNA, https://github.com/mortazavilab/PyWGCNA) [46]. After removing proteins displaying low-quality results or proteins present in some but not all 12 examined samples, the final 2103 proteins present in all four groups were inputted to pyWGCNA (S1 Data). Sample clustering indicated that the three samples of the *Tbr1$^{+/-}$*_water group clustered together and were distinct from the samples of the other groups (S1A Fig), consistent with the PCA results (Fig 2A). Based on topological matrix analysis, we identified seven major co-expression modules (Figs 2B and S1B–S1D). The "Black" module, the most significant module, exhibits the highest correlation and statistical significance (correlation 0.86, $P = 0.0$) with treatment (Fig 2B). Generally speaking, the Black module was upregulated by the supplement cocktail treatment (Fig 2C; S2 Data). The expression levels of the proteins listed in the Black module were reduced in the water-treated groups but increased in the supplement-treated groups, regardless of genotype, although *Tbr1$^{+/-}$* mice seem to present a more marked difference (Fig 2C). The entire protein list and the connectivity and relative protein levels of all 271 proteins are summarized in S2 Data.

We further investigated the individual gene ontology (GO) terms and protein networks of the Black and other modules (S2 and S3 Figs). Due to its significance, we focus here on the Black module. Synapse-related GO—including synapse organization, modulation of chemical synaptic transmission, synaptic signaling, and regulation of synapse organization—were the top GO terms associated with the Black module (Figs 2D and S4). Since we used total lysates for our proteomic analysis, the enrichment for synapse-related proteins in the top module from WGCNA supports an impact of the supplement cocktail on synaptic protein expression. In addition to synapse-related proteins, three other GO terms, i.e., generation of precursor metabolites and energy, vesicle-mediated transport, and cytoskeleton organization, were also significantly associated with the Black module (Figs 2D and S4). Proteins involved in both excitatory and inhibitory neurotransmission were also present in the Black module. Moreover, the 271 proteins in the Black module were highly connected to each other based on the STRING functional network (Fig 2D). Thus, supplement cocktail treatment may up-regulate the expression levels of these proteins to modulate synaptic activity and function and even network-level connectivity.

Note that we did not find the modules to be significantly associated with genotype (correlation < −0.8 or >0.8 and $P < 0.05$). Perhaps TBR1 primarily functions during development, influencing neuronal migration and axonal projection of the cerebral cortex and BLA, ultimately altering neuronal connectivity. In adults, TBR1 protein levels are reduced to a very low level, and its impact on gene expression becomes limited. Thus, although neuronal connections have been changed, global gene expression may not be noticeably altered in adult *Tbr1$^{+/-}$* mice.

We further picked several synaptic proteins from the list of the Black module for immunoblotting validation. We examined a total of nine proteins, including GRIA1, GRIN2B, GRIN2A, LIN7, DPYSL3, HOMER1, SAP97, SYNPO, and GABBR2 (Fig 3). The results of immunoblotting were then quantified and subjected to statistical analysis by two-way ANOVA and post-hoc testing. Among all of the examined proteins, five (i.e., GRIA1, GRIN2B, GRIN2A, DPYSL3, HOMER1) were sensitive to supplement cocktail treatment. Their protein levels after supplement cocktail treatment for 7 days were significantly increased, although the increments were only 10%–30% (Fig 3). In addition, the post-hoc test showed that GRIA1, GRIN2B, LIN7, DPYSL3, and HOMER1 presented higher protein levels in the *Tbr1$^{+/-}$*_cocktail group compared to the *Tbr1$^{+/-}$*_water group (Fig 3). Given that these proteins are critical for synaptic activity and plasticity, the upregulation of these proteins indicates that supplement cocktail treatment may improve the synaptic activity and plasticity of *Tbr1$^{+/-}$* mice by influencing the expression levels of these proteins.

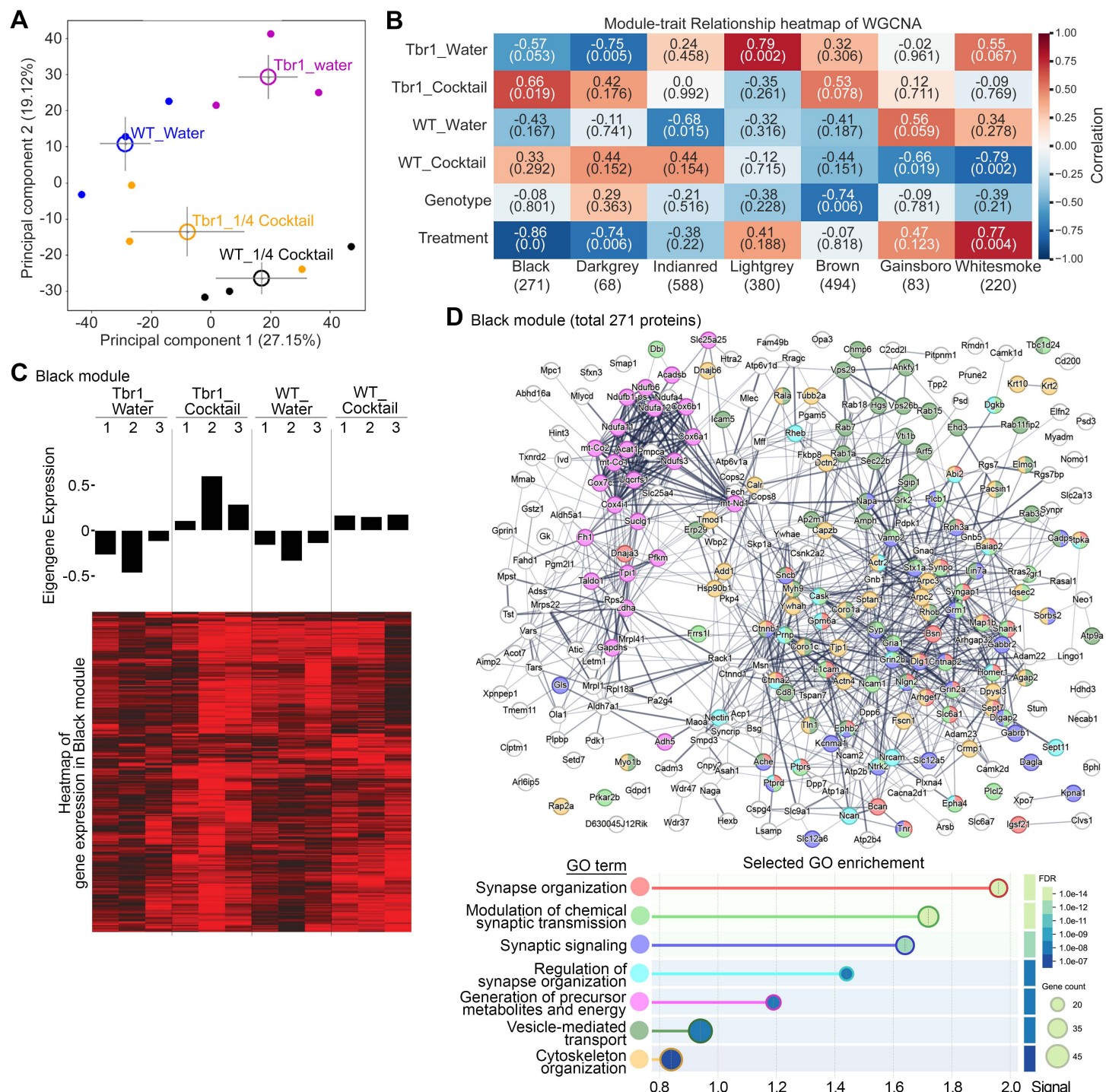

**Fig 2. Cocktail supplementation alters the total proteome of *Tbr1*^+/− mouse brains.** Four groups of total mouse brain lysates—WT_water, *Tbr1*^+/−_water, WT_1/4 cocktail, and *Tbr1*^+/−_1/4 cocktail—were subjected to LC-MS-MS analysis. The results were subjected to (A) principal component analysis (PCA) and (B–D) Python package for weighted correlation network analysis (PyWGCNA) followed by protein network analysis using STRING. (B) Heatmap of module–trait relationships identified by PyWGCNA. Seven modules and respective numbers of proteins (in brackets) are labeled below the heatmap. In addition to the four groups of samples, the same analyses were performed according to genotype and treatment. In each well, the upper numbers indicate the correlation and the lower numbers in brackets indicate the *P* value of each analysis. The analysis revealed the highest level of correlation between the "Black" module and the treatment. (C) The Eigengene expression for the Black module. Both the *Tbr1*^+/−_water and WT_water

groups display reduced expression compared to the *Tbr1*<sup>+/−</sup>_cocktail and WT_cocktail groups. (D) The protein networks and GO for the Black module were analyzed with STRING. The GO terms are color-coded. The corresponding false discovery rates and gene counts are also indicated. See also S1 and S2 Data and S1–S4 Figs.

Thus, our proteomic and bioinformatic analyses reveal that cocktail supplementation makes the protein expression profiles of *Tbr1*<sup>+/−</sup> mice more comparable to those of WT mice. In particular, the expression profiles of proteins participating in chemical synaptic transmission and mitochondrial function were altered by *Tbr1* haploinsufficiency and our cocktail treatment.

## Alteration of neural ensembles at the BLA by *Tbr1* haploinsufficiency and cocktail supplementation

To explore the impact of cocktail supplementation on neural responses in vivo, we used in vivo calcium imaging to monitor the neural activity of *Tbr1*<sup>+/−</sup> mice upon social stimulation and compared it with that of WT mice, both in the presence and absence of supplement cocktail treatment. We infected the BLA with adeno-associated virus (AAV) expressing GCaMP7c and used a miniaturized fluorescence microscope to monitor BLA activity in freely moving mice (Fig 4A). After recovery from surgery and habituation to handling by the experimenter and miniscope setup, the mice were subjected to an open field test followed by two sequential behavior tests separated by a 1-week interval. Each behavior test comprised two sessions, i.e., inanimate object exploration (OE) and reciprocal social interaction (RSI) with an unfamiliar "stranger" mouse (Fig 4A). During the 1-week interval, regular drinking water was replaced by the 1/4 cocktail supplement. Thus, the first behavior test represents the control water group (i.e., OE-1 and RSI-1), and the second one is the cocktail-treated group (i.e., OE-2 and RSI-2) (Fig 4A). After completing all experiments, the localizations of AAV infection and GRIN lens implantation were confirmed (S5A–S5C Fig). The activity and number of recorded neurons during each session were then determined based on changes in GCaMP7c signals (S5D–S5G Fig).

We used three different methods to analyze the in vivo calcium imaging results. We endeavored to determine which properties in vivo were specific to social behavior and also affected by TBR1 protein and the supplement cocktail treatment. The first approach was to evaluate the effect of *Tbr1* deficiency and cocktail supplementation on BLA neural ensembles. A previous study identified a paired neural ensemble in the BLA as driving social behaviors [47]. Similarly, determination of the within-cluster sum of squares from our data also revealed two neuronal ensembles in the BLA for all experimental groups (OE-1, OE-2, RSI-1, and RSI-2), whether WT or *Tbr1*<sup>+/−</sup> mice were being assessed (Figs 4B and S6A and S7A). Next, we determined the correlation coefficient between behaviors and neuronal ensemble activity. For OE, there was no obvious difference in correlations between behaviors and ensembles one and two for either WT or *Tbr1*<sup>+/−</sup> mice (Fig 4C, upper). In contrast, the correlation coefficient of ensemble one in RSI was noticeably higher than that of ensemble two for both WT and *Tbr1*<sup>+/−</sup> mice (Fig 4C, lower). Thus, ensemble one is more relevant to RSI, no matter the genotype. The correlations between social behavior and the activity of ensemble one also did not differ between WT and *Tbr1*<sup>+/−</sup> mice (Fig 4C, lower).

Then we analyzed the correlation of individual neuron activity with its corresponding ensemble (S6B and S7B Figs), and quantified the cumulative probability of the correlation coefficients (Fig 4D). All recorded neurons were subjected to this analysis. The stronger correlation between individual neuron activity and its corresponding ensemble indicates that those individual neurons more consistently participate in the coordinated activity of their group. We found that no matter for OE or RSI, there were differences between genotypes (i.e., *Tbr1*<sup>+/−</sup> versus WT), regardless of treatments (i.e., water versus 1/4 cocktail) (Fig 4D). The cumulative curves of *Tbr1*<sup>+/−</sup> mice were shifted right compared to WT mice, meaning that more *Tbr1*<sup>+/−</sup> neurons were more strongly correlated with their ensembles. *Tbr1*<sup>+/−</sup> and WT mice could be distinguished well based on these correlation coefficients (Fig 4D). However, this alteration was not specific to social behavior and it was not improved by supplement cocktail treatment.

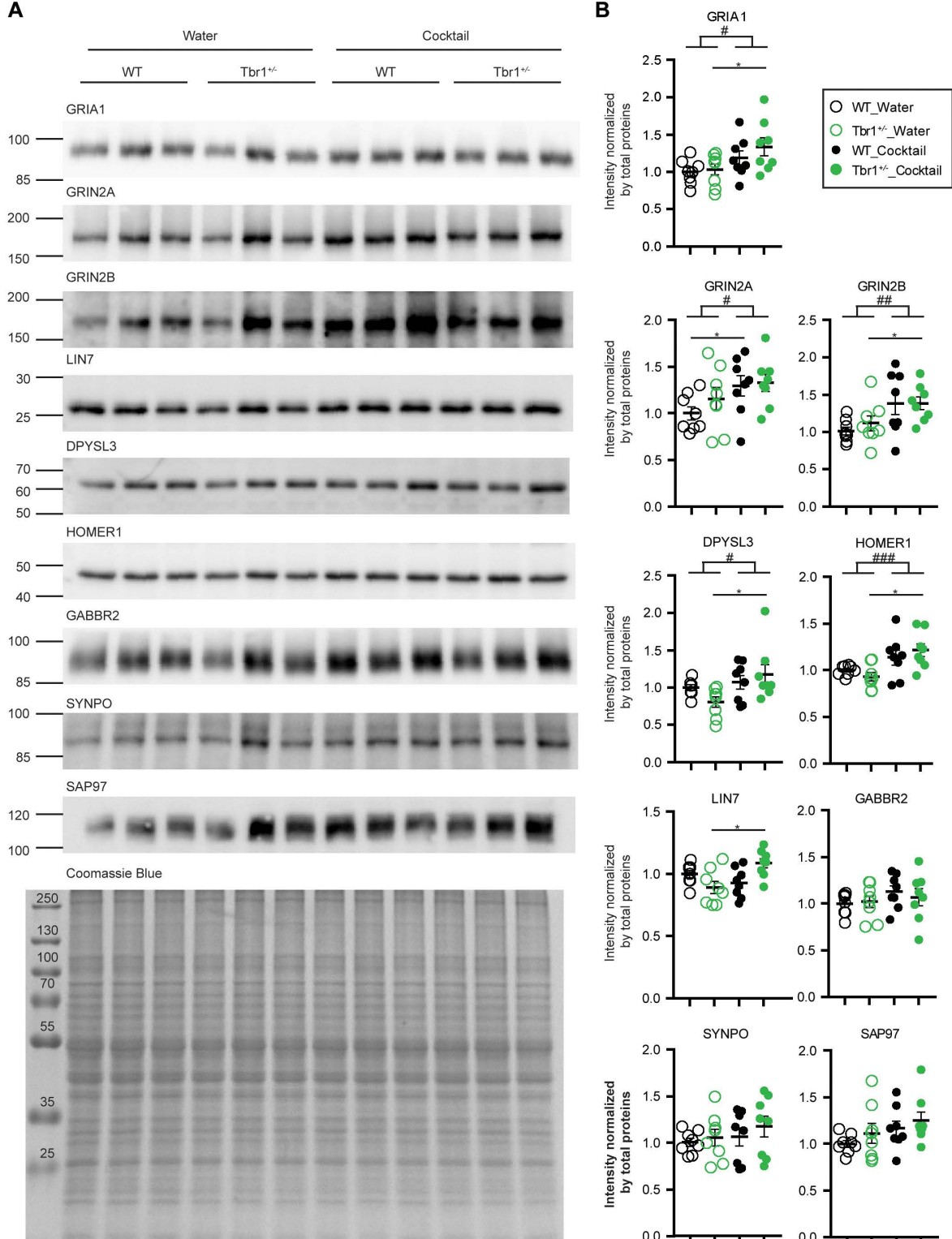

**Fig 3. Immunoblotting validation of proteins listed in the Black module.** Four groups of total mouse brain lysates—WT_water, *Tbr1⁺/⁻*_water, WT_1/4 cocktail, and *Tbr1⁺/⁻*_1/4 cocktail—were prepared for immunoblotting. Each group comprises eight mice for analysis. A total of nine antibodies were applied for immunoblotting, as indicated. Quantification was performed by normalization against the total protein levels revealed by Coomassie

Blue stain. (A) Representative immunoblots. (B) Quantification of immunoblots. The statistical analysis was conducted using two-way ANOVA and Bonferroni's post-hoc correction. # $P < 0.05$; ## $P < 0.01$; ### $P < 0.001$ in two-way ANOVA test. *, $P < 0.05$; **, $P < 0.01$ in the post-hoc test. Full scans of immunoblots are available in S1 Raw Images. The data underlying the graphs in (B) can be found in the S3 Data. All statistical analyses and results, including the actual $P$-values, are summarized in S4 Data.

In addition to analyzing all recorded neurons (Fig 4D), we analyzed dual-recorded neurons in both OE-1/OE-2 and RSI-1/RSI-2, with a view to revealing the specific response of a particular neuron. We recorded a total of 300 responsive neurons in four WT mice and 242 such neurons in four *Tbr1*$^{+/-}$ mice: 27 (9%) and 12 (4.9%) dual-recorded neurons for OE in the WT and *Tbr1*$^{+/-}$ mice, respectively; and 49 (16.3%) and 16 (6.6%) dual-recorded neurons for RSI in the WT and *Tbr1*$^{+/-}$ mice, respectively. Thus, *Tbr1*$^{+/-}$ mice had fewer dual-recorded neurons than WT mice in both the OE and RSI contexts (Fig 4E and 4F).

Among these dual-recorded neurons, WT neurons were positively correlated between OE-1 and OE-2, whereas *Tbr1*$^{+/-}$ neurons were negatively correlated, although in neither case was the correlation statistically significant (Fig 4E, WT, $p = 0.1518$; *Tbr1*$^{+/-}$, $p = 0.3883$). The correlation coefficients for individual WT neurons were widely distributed from values of zero to one. However, the correlation coefficients for *Tbr1*$^{+/-}$ neurons were enriched in the range of 0.3 to 0.8 (Fig 4E). Thus, there were fewer repetitively responsive neurons in *Tbr1*$^{+/-}$ mice in the OE condition compared to WT mice and they were less diverse.

In terms of RSI, though the correlations between the activities of individual WT neurons and their ensemble were still widely distributed from values of zero to one for both the RSI-1 and RSI-2 groups, we detected a significant positive correlation between RSI-1 and RSI-2 (Fig 4F, left, $p = 0.0154$). This outcome indicates that although the correlation is heterogeneous, the activity correlation of a particular neuron with the ensemble tended to be maintained between RSI-1 and RSI-2 in WT mice. In contrast, there was no correlation between RSI-1 and RSI-2 for *Tbr1*$^{+/-}$ mice (Fig 4F, right, $p = 0.4041$). Moreover, the correlation coefficients of individual neurons in RSI-1 were widely distributed from values of 0.2 to 1, but with coefficients enriched in the range of 0.4–0.8 for RSI-2 (Fig 4F, right). Thus, cocktail supplementation alters the activity correlation of repetitively responsive neurons in *Tbr1*$^{+/-}$ but not WT mice during RSI.

Consequently, *Tbr1* haploinsufficiency results in a stronger correlation between individual BLA neuron activity and its corresponding ensemble. Moreover, cocktail supplementation alters the activity correlation of repetitively responsive neurons in the *Tbr1*$^{+/-}$ BLA.

## Activity change of BLA neurons by *Tbr1* haploinsufficiency and supplement cocktails

Our second set of calcium imaging analyses aimed to investigate the responses of individual neurons during behaviors. We identified three types of neurons based on correlations of their activities with behavior [48], i.e., positively correlated, negatively correlated, and irrelevant neurons (Fig 5A and 5B). Neurons displaying a positive correlation reflected those with a higher probability of being activated during behavior. Those exhibiting a negative correlation tended to be activated between two behavioral bouts. Irrelevant neurons did not present a clear correlation (positive or negative) with behavior (Fig 5A and 5B). Among a total of 300 and 242 activated neurons recorded from WT and *Tbr1*$^{+/-}$ mice, respectively, the majority (>80%) of activated BLA neurons were irrelevant to OE and RSI for both WT and *Tbr1*$^{+/-}$ mice (Fig 5C, gray area). During both the OE-1 and OE-2 tests, approximately 4% and 1% of activated cells were defined as positively or negatively correlated neurons, regardless of genotype (Fig 5C), indicating that *Tbr1*$^{+/-}$ mice do not differ from WT mice in terms of numbers of activated neurons during OE.

In terms of RSI, the percentages of positively- and negatively-correlated neurons in WT mice were 9.3% and 3% for RSI-1, respectively, which rose to 12.3% and 6.3% for RSI-2, i.e., following 1 week of cocktail treatment (Fig 5C). Thus, cocktail supplementation of WT mice increased the number of both positively- and negatively-correlated neurons in the BLA. For *Tbr1*$^{+/-}$ mice, positively correlated neurons accounted for 11.6% of total recorded neurons in RSI-1. However,

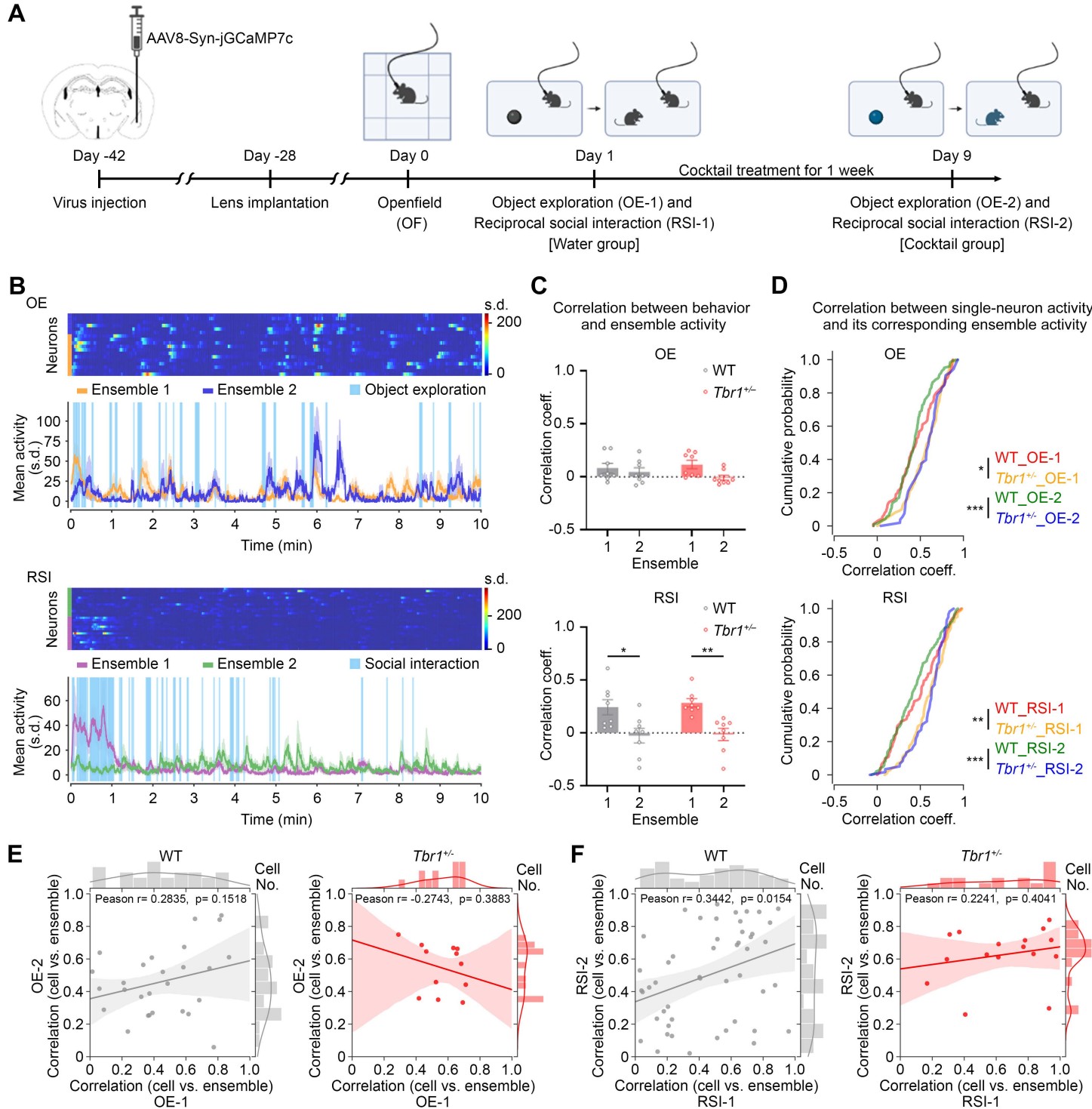

**Fig 4. Neural ensembles in the BLA are influenced by *Tbr1* haploinsufficiency and cocktail supplementation.** (A) Schematic of the experimental design (WT: *n* = 4; *Tbr1⁺/⁻*: *n* = 4). (B) Activity of neurons in the basolateral amygdala (BLA) during object exploration (OE, top) and reciprocal social inter-action (RSI, bottom) tests. A set of representative results, including individual neuron activity and mean activity for each neuronal ensemble, is shown. (C) The correlation between the mean neuronal ensemble activity and behavior (binary vector) in the OE (top) and RSI (bottom) tests. (D) Correlation of the activity of all recorded neurons and their corresponding ensembles. The cumulative probabilities of the correlation coefficient of WT and *Tbr1⁺/⁻* mice in OE-1, OE-2, RSI-1, and RSI-2 are shown. (E, F) Activity correlations between dual-recorded neurons and their corresponding neural ensembles. The

data points represent the individual neurons. (E) Object exploration, $n = 27$ from four WT mice; $n = 12$ from four $Tbr1^{+/-}$ mice. (F) RSI, $n = 49$ from four WT mice; $n = 16$ from four $Tbr1^{+/-}$ mice. Data in (C) is mean ± SEM. Statistical analysis in (C) by two-way ANOVA followed by Bonferroni post hoc test. The Kolmogorov–Smirnov test was used for cumulative probabilities in (D). * $P < 0.05$, ** $P < 0.01$, *** $P < 0.001$. See also S5–S7 Figs. The data underlying the figure can be found in the S3 Data. All statistical analyses and results, including the actual $P$-values, are summarized in S4 Data. *The figure was created in BioRender. Lin, M. (2025)* https://BioRender.com/cyi56kh.

we did not identify any negatively correlated neurons in the $Tbr1^{+/-}$ RSI-1 group (Fig 5C). In the RSI-2 group, negatively correlated neurons were present (1.2%) in $Tbr1^{+/-}$ mice, but the percentage of positively correlated neurons declined to 5% (Fig 5C, RSI-2 group). Thus, these results suggest the differential neuronal response of $Tbr1^{+/-}$ and WT mice to social stimulation as well as the differing effects of cocktail supplementation on $Tbr1^{+/-}$ and WT mice.

Next, we determined the mean activity of positively correlated neurons during behaviors. The amplitudes of mean activity were comparable among the OE-1 and OE-2 groups of WT and $Tbr1^{+/-}$ mice (Fig 5D, upper). However, the mean activity of the $Tbr1^{+/-}$ RSI-1 group was noticeably higher than that of the WT RSI-1 group (Fig 5D, lower). Moreover, cocktail supplementation reduced the mean activity of $Tbr1^{+/-}$ neurons in the RSI-2 group (Fig 5D, lower). These results imply that $Tbr1$ deficiency results in hyperactivation of positively correlated neurons in the BLA during social interaction and that the cocktail treatment can correct this hyperactivity of $Tbr1^{+/-}$ BLA neurons.

We further determined how neurons were repetitively activated in these different behavior tests (Fig 5E and 5F). First, we used concentric circles to summarize the properties of each recorded BLA neuron in WT and $Tbr1^{+/-}$ mice (Fig 5E). Based on behavioral tests and treatments, the concentric circles were then separated into four parts, i.e., OE-1, RSI-1, OE-2, and RSI-2 (Fig 5E), with the same responsive neurons being connected by lines crossing the central area of the circles. We found that there were more connecting lines for WT mice compared to $Tbr1^{+/-}$ mice (Fig 5E), implying that more responsive BLA neurons were shared among the tests in WT mice, although those shared neurons still represented a minority of total BLA neurons (Fig 5E and 5F). In particular, only seven neurons (i.e., 8.1% of responsive neurons) of WT mice and one neuron (i.e., 2.4% of responsive neurons) of $Tbr1^{+/-}$ mice were shared by RSI-1 and RSI-2 (Fig 5F, right). These results indicate that different BLA neurons respond to the same types of behavior at different times, consistent with a previous observation [47]. In $Tbr1^{+/-}$ mice, the beneficial effect of cocktail supplementation on RSI-2 is mainly mediated by neurons that differ from those that are responsive during RSI-1. Accordingly, our supplement cocktail appears to influence the responsiveness of individual neurons and consequently alters their connectivity to improve the responses of mice to social stimulation.

## Cocktail treatment alters the BLA neural network

As a final approach to our in vivo calcium imaging analysis, we characterized the functional neural network of the BLA in our mice. When we examined the individual responses of BLA neurons, we noticed that the positively correlated neurons of the $Tbr1^{+/-}$ RSI-1 group tended to display higher activity at the beginning of the RSI test and their activities seemed more correlated with each other (Fig 6A). However, this property disappeared in RSI-2, i.e., after cocktail supplementation. Moreover, this synchronized activity at the beginning of the RSI-1 session was not observed for WT mice (Fig 6A). To analyze this property, we measured degree centrality, which represents a given neuron's connection with others based on the similarity of their activation patterns [49,50]. All recorded neurons, including positive, negative, and irrelevant neurons, were subjected to this analysis.

For RSI-1, all recorded neurons in $Tbr1^{+/-}$ mice were tightly clustered in the respective network graph (Figs 6B and S8) and had a higher degree centrality compared to that of WT mice (Fig 6C). When we separated sociality-linked neurons (encompassing both positively and negatively correlated neurons) and irrelevant neurons, the difference in degree centrality between $Tbr1^{+/-}$ and WT mice was more obvious for sociality-linked neurons (Fig 6C, middle panel) and it was absent for irrelevant neurons (Fig 6C, right panel). Thus, the difference observed for all neurons is mainly attributable to

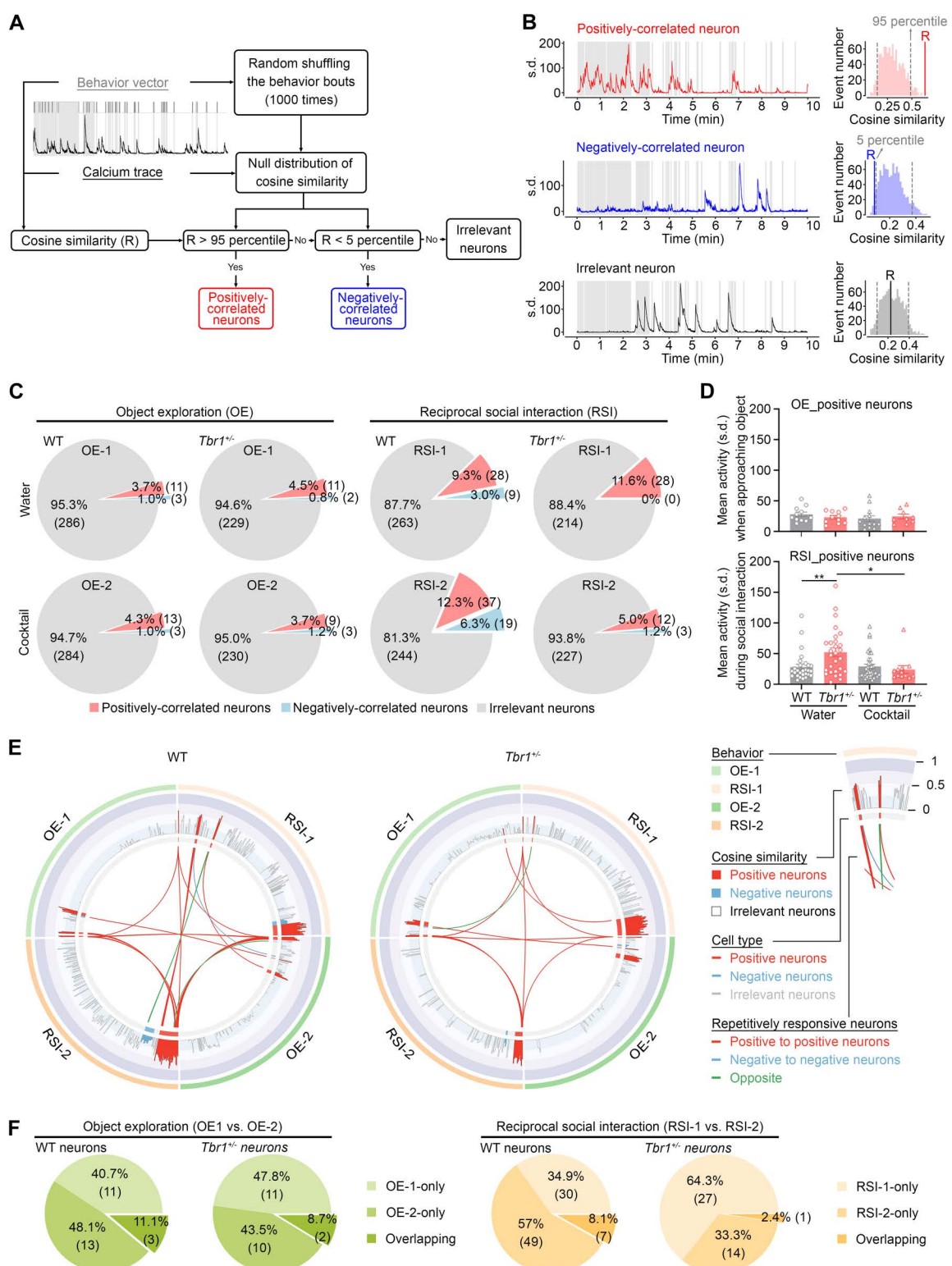

**Fig 5. Populations and responses of sociality-linked neurons in the BLA are influenced by *Tbr1* deficiency and cocktail supplementation.**
(A) Definition of the behavior-related neurons. (B) Representative in vivo calcium imaging traces of behavior-related positively correlated (top), negatively correlated (middle), and irrelevant (bottom) neurons. Left, neuronal activity during the entire session. Right, comparison between the raw cosine

similarity (R) and null distribution of shuffled data. (C) Ratio of object exploration- and sociality-linked neurons (*n* = 300 from four WT mice; *n* = 242 from four *Tbr1*⁺ᐟ⁻ mice). (D) Mean activity of object exploration- and sociality-linked neurons during behavior tests. Only positively correlated neurons were analyzed. (E) Connectrograms to reveal repetitively recorded neurons across the four behavioral sessions. (F) Overlap between the water and cocktail experimental groups in terms of object exploration- and sociality-linked neurons. Data in (D) represents mean ± SEM. * *P* < 0.05, ** *P* < 0.01; two-way ANOVA with Bonferroni post hoc test. The data underlying the figure can be found in the S3 Data. All statistical analyses and results, including the actual *P*-values, are summarized in S4 Data.

sociality-linked neurons. This result also indicates that *Tbr1* deficiency prompts enhanced functional connectivity among BLA neurons.

Then we investigated the effect of cocktail supplementation, i.e., the response of RSI-2. The degree centrality of all neurons was not affected ([Fig 6C], left). However, the degree centrality of *Tbr1*⁺ᐟ⁻ sociality-linked neurons was reduced by the cocktail treatment to levels comparable to those of WT mice ([Fig 6C], middle). Irrelevant neurons exhibited a different response, with WT irrelevant neurons displaying reduced degree centrality after cocktail supplementation, resulting in their being different to the irrelevant neurons of *Tbr1*⁺ᐟ⁻ mice ([Fig 6C]). Nevertheless, these results indicate that cocktail supplementation corrects the abnormal functional connectivity of sociality-linked BLA neurons in *Tbr1*⁺ᐟ⁻ mice.

In addition to RSI, we also analyzed the BLA functional network for OE ([S9 Fig]). In contrast to RSI, cocktail supplementation did not alter the degree centrality of *Tbr1*⁺ᐟ⁻ neurons during this behavior test, no matter if all recorded neurons, neurons associated solely with OE, or irrelevant neurons were assessed ([Fig 6D] and [6E], *Tbr1*⁺ᐟ⁻ OE-1 versus OE-2). The only noticeable differences we detected were between WT and *Tbr1*⁺ᐟ⁻ neurons ([Fig 6D] and [6E], all neurons of WT versus *Tbr1*⁺ᐟ⁻ in OE-2 and irrelevant neurons of OE-1). These results strengthen the specific roles of TBR1 and cocktail supplementation in regulating the BLA's functional network in response to social stimulation.

Overall, the results of our suite of in vivo calcium imaging analyses indicate that *Tbr1* deficiency results in hyperactivity and hyperconnectivity of BLA neurons during social interaction, consequently modulating BLA neuronal ensembles. Treatment with our supplement cocktail corrects the abnormal neuronal activity and connectivity of the BLA in *Tbr1*⁺ᐟ⁻ mice.

### Synergistic effects of dietary supplements on social behaviors

Next, we investigated the effect of the cocktail supplementation on mouse behaviors. First, we confirmed the synergistic effect of the supplement cocktail on RSI. *Tbr1*⁺ᐟ⁻ mice were subjected to five sequential RSI tests with a 1-week interval between tests starting from postnatal day 60 (P60). The first RSI test was the water control and the second to fifth tests were to investigate sequentially the effects of one-week administration with 0.45% BCAA, 1% serine, 20 ppm zinc, or 1/4 cocktail ([Fig 7A]). Supplementation with individual nutrients or 1/4 cocktail did not influence the social behaviors of WT mice ([Fig 7B], left). For *Tbr1*⁺ᐟ⁻ mice, administrations of 0.45% BCAA, 1% serine, or 20 ppm zinc alone did not enhance their social behaviors in RSI. However, the 1/4 cocktail treatment improved the social behaviors of the *Tbr1*⁺ᐟ⁻ mice in the fifth RSI test ([Fig 7B], middle). When we compared the responses of the WT and *Tbr1*⁺ᐟ⁻ mice, we detected differences in the water and individual supplement assay groups, but not for the cocktail-provisioned group ([Fig 7B], right), supporting the synergistic effect of our supplement cocktail on social behaviors.

To further consolidate the synergistic effect of our supplement cocktail, we investigated another ASD mouse model, i.e., *Cttnbp2*⁺ᐟᴹ¹²⁰ᴵ mice. We have demonstrated previously that 0.45% BCAA or 40 ppm zinc supplemented in drinking water is sufficient to improve the social behaviors of *Cttnbp2*⁺ᐟᴹ¹²⁰ᴵ mice [13,14,18]. Therefore, we investigated the response of *Cttnbp2*⁺ᐟᴹ¹²⁰ᴵ mice to a "1/8 cocktail", in which the concentration of BCAA had been further reduced to one eighth the original dose (i.e., 0.225%), the amount of serine was reduced to one-quarter of the original (i.e., 0.5%), and the amount of zinc was maintained at 20 ppm ([Fig 7C]). Based on the results of two consecutive RSI tests (one with water, the other with nutrient supplement), we did not detect a noticeable improvement in the social behaviors of *Cttnbp2*⁺ᐟᴹ¹²⁰ᴵ mice that were individually supplemented with 20 ppm zinc, 0.225% BCAA, or 0.5% serine ([Fig 7D]). Importantly, supplementation with

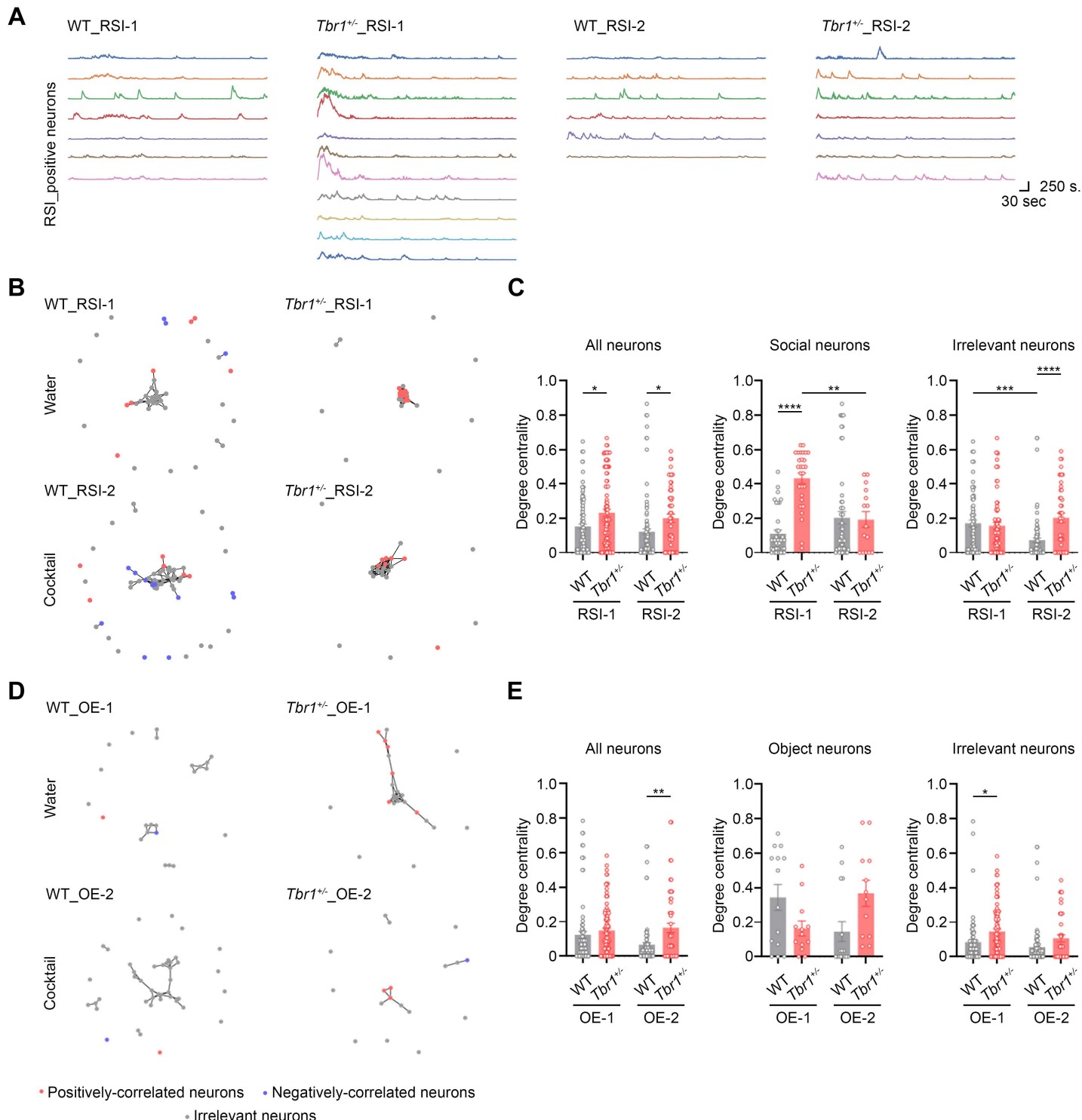

**Fig 6. Cocktail supplementation increases the diversity of activation patterns in the sociality-linked neurons of *Tbr1*$^{+/-}$ mice.** (A) Representative neuronal activity patterns of RSI-positive neurons in WT and *Tbr1*$^{+/-}$ mice. (B) Functional networks of WT and *Tbr1*$^{+/-}$ mice in RSI. Links (lines) between the nodes represent significant similarity in activation patterns relative to shuffled data. (C) Degree centrality of neurons in WT and *Tbr1*$^{+/-}$ mice across the RSI-1 and RSI-2 sessions. (D) Functional networks of WT and *Tbr1*$^{+/-}$ mice in OE. Links (lines) between the nodes represent significant similarity in activation patterns relative to shuffled data. (E) Degree centrality of neurons in WT and *Tbr1*$^{+/-}$ mice across the OE-1 and OE-2 sessions. The results

of WT mouse #3 and *Tbr1+/−* mouse #4 are presented as examples. Data in (C) and (E) are presented as means±SEM. The data points of individual neurons are also shown. Two-way ANOVA with Bonferroni post hoc test was used for statistical analysis. ** $P < 0.01$; *** $P < 0.001$; **** $P < 0.0001$. See also S8 and S9 Figs. The data underlying the figure can be found in the S3 Data. All statistical analyses and results, including the actual $P$-values, are summarized in S4 Data.

the 1/8 cocktail enhanced the social interaction of *Cttnbp2+/M120I* mice with unfamiliar mice in RSI (Fig 7D). Thus, mixing low-dose zinc, BCAA, and serine supplements results in a synergistic effect that improves the social behaviors of multiple autism mouse models.

## Supplement cocktails improve ASD-associated behaviors of multiple mouse models

Apart from a 1-week treatment regimen, we further investigated the long-term effects of the 1/4 cocktail on *Tbr1+/−* mice. Supplementation starting from weaning and it continued until all experiments had been completed. We performed a

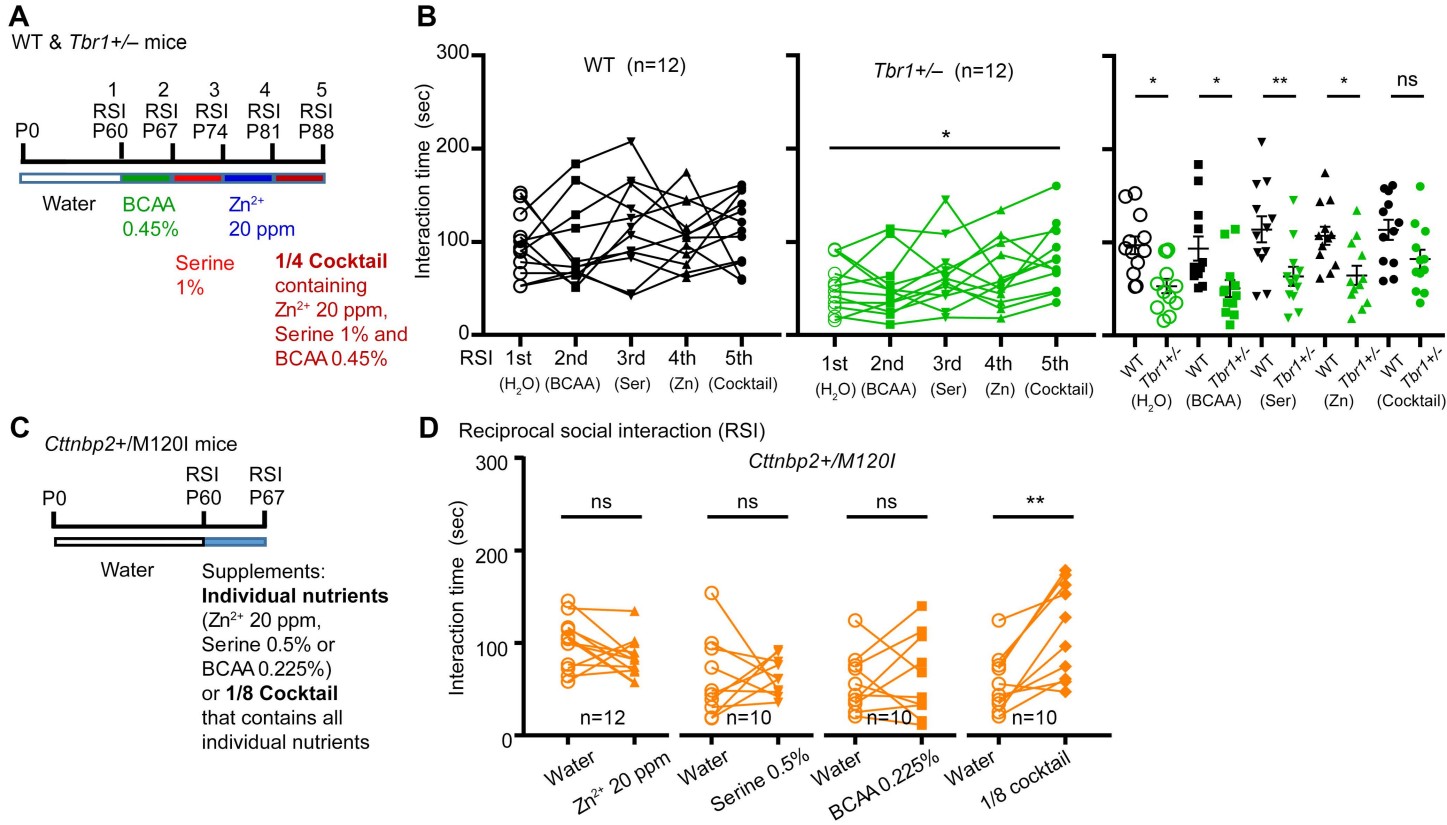

**Fig 7. Mixing low-dose nutrients synergistically improves social behaviors of autism mouse models.** (A, C) Experimental design of dietary supplementation and behavior assays. RSI, reciprocal social interaction. (B) Synergistic effects of the 1/4 cocktail on RSI of WT and *Tbr1+/−* mice. Individual nutrients, including BCAA (0.45%), serine (1%), and $Zn^{2+}$ (20 ppm), and the 1/4 cocktail were sequentially applied to mice for 1 week for each supplementation assay. Five RSI tests were conducted as indicated to investigate the effect of supplements on RSI. Left: the response of WT mice in each test; Middle: the response of *Tbr1+/−* mice in each test; Right: the comparison of WT and *Tbr1+/−* mice in each test. (D) Synergistic effects of the 1/8 cocktail on *Cttnbp2+/M120I* mice. Different from (A) and (B), each *Cttnbp2+/M120I* mouse was only subjected to two RSI tests: one is water control; the other is supplement treatment. Data are presented as data points for individual mice and/or means±SEM. The numbers (*n*) of examined mice for each group are indicated. (B) Left and Middle: paired *t* test was performed to examine the difference between water control and other experimental groups, individually. Right: Two-way ANOVA with Bonferroni post hoc test. (D) Paired *t* test. * $P < 0.05$, ** $P < 0.01$, ns = non-significant. The data underlying the figure can be found in the S3 Data. All statistical analyses and results, including the actual $P$-values, are summarized in S4 Data.

sequential series of behavioral assays, including open field, elevated plus maze, RSI, three-chamber test and cued fear conditioning, starting from P60 (Fig 8A). Long-term supplementation of the 1/4 cocktail did not noticeably affect mouse body weight (Fig 8B). Compared to the control group that drank regular water, neither *Tbr1*$^{+/-}$ nor WT mice administered the 1/4 cocktail exhibited a difference in behavior in the open field or elevated plus maze assays (Fig 8C and 8D), consistent with our previous observation that *Tbr1* haploinsufficiency does not influence locomotion or anxiety [39].

For auditory fear conditioning, similar to results reported previously [39], the freezing response of *Tbr1*$^{+/-}$ mice that drank regular water was lower than that of their WT littermates (Fig 8E). However, the performance of *Tbr1*$^{+/-}$ mice provisioned with the 1/4 cocktail supplement was comparable to that of their WT littermates (Fig 8E). Thus, the 1/4 cocktail treatment improves the memory performance of *Tbr1*$^{+/-}$ mice to a level comparable to that of WT littermates.

In terms of RSI, long-term 1/4 cocktail treatment also increased the interaction time of *Tbr1*$^{+/-}$ mice with unfamiliar (stranger) mice (Fig 8F). This effect was specific to the mutant mice because the 1/4 cocktail did not alter the performance of WT littermates in RSI (Fig 8F). Similarly, the 1/4 cocktail also improved the sociability of *Tbr1*$^{+/-}$ mice in the three-chamber test, as reflected by a longer interaction time with stranger one compared to an inanimate object, as well as the comparable preference index between cocktail-supplemented *Tbr1*$^{+/-}$ mice and WT mice (Fig 8G, left). Consistent with a previous finding that *Tbr1* deficiency is not involved in novelty preference of social interaction [39], we also did not detect a difference in novelty preference regardless of genotype or treatment (Fig 8G, right). Thus, long-term 1/4 cocktail treatment improves the social deficits caused by *Tbr1* haploinsufficiency without noticeable side effects on body weight, locomotion or anxiety.

Apart from *Tbr1*$^{+/-}$ and *Cttnbp2*$^{+/M120I}$ mice, we further analyzed the effects of a supplement cocktail on yet another ASD mouse model, i.e., *Nf1*$^{+/-}$ mice. Our previous study showed that 0.9% Leu in drinking water did not improve the social behavior of *Nf1*$^{+/-}$ mice [16]. Therefore, we designed a "1/2 cocktail", in which the concentrations of BCAA, L-serine, and zinc were all reduced to half the original dose. To monitor the behavioral changes of the same mice, we subjected them to two consecutive tests of RSI with a one-week interval, providing them with the 1/2 cocktail during the interval (Fig 8H). We found that the *Nf1*$^{+/-}$ mice exhibited longer interaction times with unfamiliar mice in the second RSI compared to the first RSI (Fig 8I). *Tbr1*$^{+/-}$ mice and *Cttnbp2*$^{+/M120I}$ mice were both included in this assay as positive controls (Fig 8I). These results support that our 1/2 cocktail exerts a beneficial effect in terms of the social behavior of multiple ASD mouse models.

## Discussion

Herein, we have demonstrated the abnormal synaptic proteomes, as well as hyperactivity and hyperconnectivity of BLA neurons, in *Tbr1*$^{+/-}$ mice. Using this model, we have revealed the effects of a supplement cocktail containing BCAA, serine, and zinc in correcting the altered proteomes and neural ensembles of *Tbr1*$^{+/-}$ mice. We further applied mouse behavioral assays to reveal the synergistic effects of mixing low-dose individual supplements on multiple ASD mouse models, including *Tbr1*$^{+/-}$, *Nf1*$^{+/-}$ and *Cttnbp2*$^{+/M120I}$ mice. Thus, dietary supplements that enhance synaptic activity and protein synthesis (Fig 1) can correct the abnormal neural activation and connectivity and improve the social behaviors of multiple ASD mouse models. Such supplement cocktails could represent a safe and available treatment for various ASD conditions.

For unknown reasons, the BLA is the region most susceptible to *Tbr1* haploinsufficiency [39]. Reduction or absence of the posterior part of the anterior commissure, a white matter structure linking the two BLA in the two brain hemispheres, is an evolutionarily conserved feature shared by humans and mice displaying monoallelic mutation or deletion in the *TBR1* gene [39,51,52]. Our previous study further demonstrated that an increase in BLA activity upon treatment with D-cycloserine, an analog of D-serine, improved the social interaction behavior of *Tbr1*$^{+/-}$ mice [39]. Recently, we performed a whole-brain analysis of C-FOS expression, revealing that *Tbr1* haploinsufficiency globally enhances the neural activity of the entire brain in the absence of particular stimulation. However, the correlated neuronal activity between the BLA and other brain regions

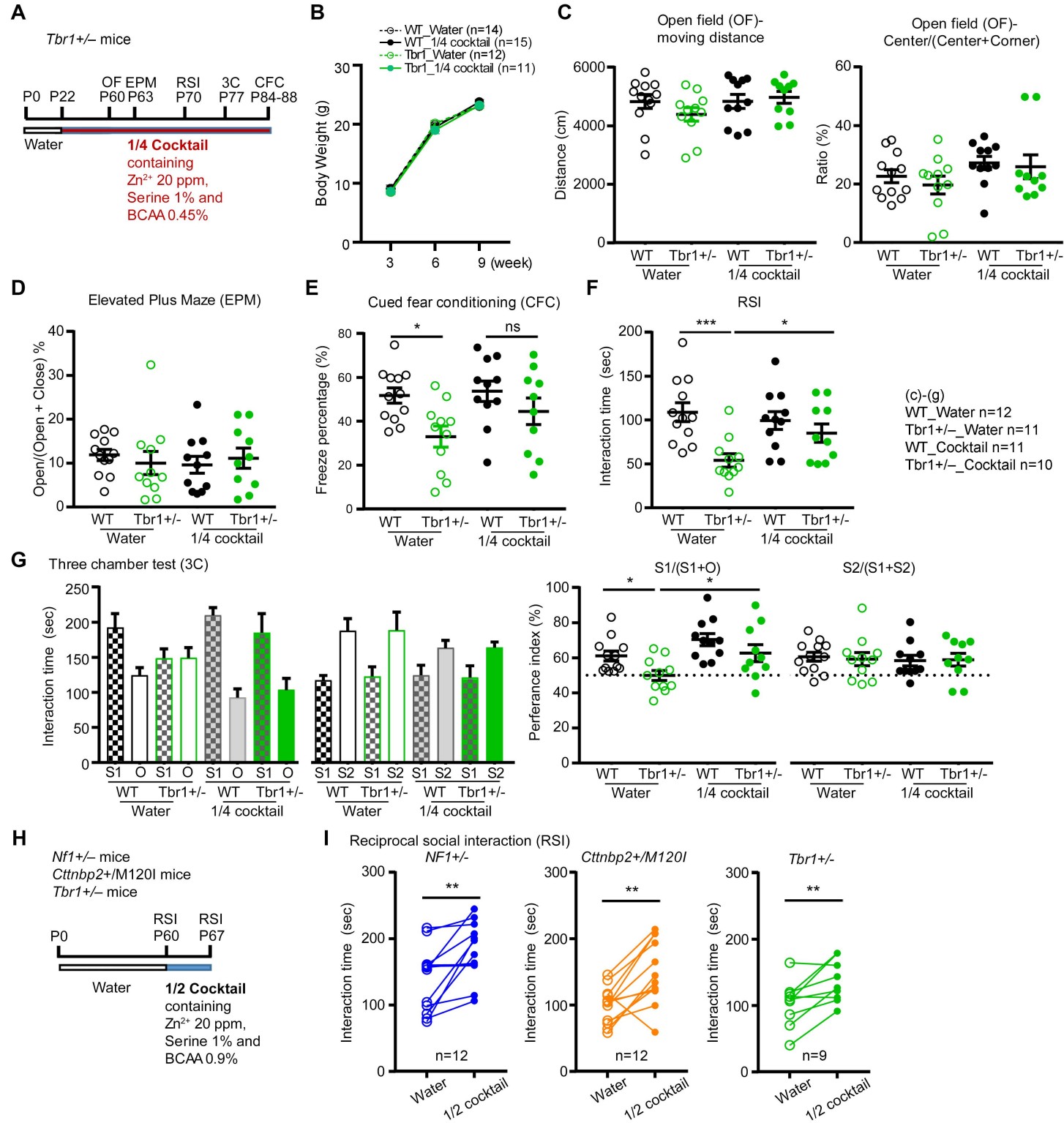

**Fig 8. Cocktail supplementation improves the social behaviors of three autism mouse models.** (A, H) Experimental design of cocktail supplementation and behavior assays. OF, open field; EPM, elevated plus maze; 3C, three chamber test; RSI, reciprocal social interaction; CFC, cued fear conditioning. (B-G) The effect of long-term 1/4 cocktail treatment on *Tbr1+/−* mice. (B) Body weight. (C) Open field. (D) Elevated plus maze. (E) Cued fear

conditioning. (F) Reciprocal social interaction. (G) Three-chamber test. (I) The social behaviors of *NF1*[+/−], *Cttnbp2*[+/M120I], and *Tbr1*[+/−] mice in the RSI assay were improved by 1/2 cocktail treatment. Data are presented as data points for individual mice and/or means ± SEM. The numbers (*n*) of examined mice for each group are indicated. (B, I) Paired *t* test. (C–G) Two-way ANOVA with Bonferroni post hoc test. * $P < 0.05$, ** $P < 0.01$, ns = non-significant. Note there is no significant difference in (B–D), though no labeling is indicated. The data underlying the the figure can be found in the S3 Data. All statistical analyses and results, including the actual *P*-values, are summarized in S4 Data.

is reduced in the *Tbr1*[+/−] mice, indicating that the degree of whole-brain synchronization, namely connectivity, of the BLA is reduced by *Tbr1* haploinsufficiency [43]. Importantly, theta-burst stimulation at the BLA was found to enhance whole-brain synchronization of *Tbr1*[+/−] brains to a level comparable to that of WT mice and improve nose-to-nose interactions (i.e., sociality) of *Tbr1*[+/−] mice [43]. In the current study, we found that *Tbr1* haploinsufficiency results in a stronger correlation among sociality-linked BLA neurons in response to social stimulation, especially at the beginning of the behavioral test. Thus, *Tbr1* haploinsufficiency also induces BLA neuron hyperactivation, resulting in stronger correlations among BLA neurons upon social stimulation, which may be relevant to the alteration of synaptic proteomes caused by *Tbr1* haploinsufficiency.

Although the concentrations of individual supplements in our cocktails are low, mixing them elicited a beneficial effect. Our LC–MS–MS and bioinformatic analyses demonstrate that a combination of individual supplements can influence synaptic and metabolic proteomes, shifting mutant neurons toward a status comparable to WT neurons. The in vivo calcium imaging analyses also reveal that cocktail supplementation corrects the abnormal hyperactivity and synchronization of BLA neurons in *Tbr1*[+/−] mice in response to social stimulation, indicating that the supplement cocktail has reset the connectivity of BLA neurons, thereby promoting appropriate responses to social stimulation. It would be intriguing to investigate in the future if whole-brain connectivity can also be corrected by dietary supplementation.

Consistent with a previous study [47], we found that very few neurons were repetitively responsive to the first and second social stimulations. Given that no negatively correlated neuron was present in RSI-1, *Tbr1* haploinsufficiency alters the cell type responding to social stimulation. This change in cell type likely contributes to the altered neural connectivity and synchronization we observed in *Tbr1*[+/−] mouse brains. However, it remains elusive how *Tbr1* haploinsufficiency changes the responsiveness of different cell types in BLA. Dissecting the cell types by means of snRNA-seq analysis would likely reveal the impact of *Tbr1* haploinsufficiency on BLA cell types. Our observation that the supplement cocktail reduced the number of positively correlated neurons but increased the negatively correlated neurons in *Tbr1*[+/−] mice is also interesting. Accordingly, the absolute number of responsive neurons in the BLA may not be the most critical factor in controlling social behaviors. Instead, appropriate connectivity in the BLA is likely more important. How *Tbr1* haploinsufficiency and nutrient supplementation alter cell types and modulate the neuronal response, and ultimately influence connectivity, is an intriguing topic that warrants further exploration.

Although the three ASD-linked genes *Tbr1*, *Nf1*, and *Cttnbp2* exert distinct molecular roles, they all influence neuronal development and synaptic functions [38,44,45,53]. Herein, we show that the social behaviors of these three mouse models are improved by supplement cocktails containing zinc, BCAA, and serine, all of which regulate synapse formation and signaling, albeit via distinct mechanisms (Fig 1). However, we did uncover varying sensitivities of the different mutant mouse lines to the supplement cocktails. For instance, the 1/2 cocktail improved social interaction of all of the examined ASD models. For *Tbr1*[+/−] mice, the 1/4 cocktail was sufficient, whereas the 1/8 cocktail comprising the lowest dosages was effective for *Cttnbp2*[+/M120I] mice. We speculate that these different sensitivities may be relevant to the function and/or phenotypic severity of the ASD mutations. *Tbr1* haploinsufficiency disrupts axonal projection of the BLA and results in a reduction or absence of the anterior commissure in both humans and rodents. These neuroanatomical deficits occur early during development and are exhibited by all individuals possessing *TBR1* mutation [39,51,52]. Similar to *Tbr1*[+/−] mice, *Cttnbp2*[+/M120I] mice also exhibit typical ASD-linked behavioral deficits. However, unlike *Tbr1*[+/−] mice, *Cttnbp2*[+/M120I] mice do not have recognizable neuroanatomical deficits [14,26]. When ASD mutation causes structural defects, it would require a higher dose of supplement cocktails to rescue the behavioral deficits.

Based on previous proof-of-concept studies [13,14,16,22], dietary therapy has to be conducted continuously to maintain the beneficial effects. For ASD treatment, it is better to perform the intervention as early as possible. Therefore, the timescale for dietary therapy of ASD is the entire life span. It is certainly prudent to minimize all potential side effects caused by long-term high-dose supplementation of a single nutrient. Although zinc is an essential mineral, oral high-dose zinc supplementation may cause diarrhea, dizziness, headache, nausea, upset stomach, and vomiting (Mayo Clinic, https://www.mayoclinic.org/drugs-supplements-zinc/art-20366112). Both zinc and copper are divalent metal ions. Long-term high-dose zinc supplementation can interfere with copper absorption and lead to copper deficiency [54,55]. For amino acids, single amino acid supplementation is generally not recommended because of the negative impact on nitrogen balance. It may reduce metabolic efficiency and result in kidney problems [56,57]. Due to the imbalance of amino acids, single amino acid supplementation may also cause growth problems in children. Moreover, the uptake and transportation of amino acids require specific transporters located on the plasma membrane. BCAA and other aromatic amino acids (including tryptophan, tyrosine, and phenylalanine) share the same transporter [33]. Thus, these two subtypes of amino acids compete with each other for transportation. Long-term high-dose supplementation of BCAA may influence brain function by reducing aromatic amino acid transport at the blood–brain barrier. Since aromatic amino acids are the precursors for critical neurotransmitters, such as serotonin and catecholamines, reducing the concentration of aromatic amino acids can have functional consequences, such as altering hormonal function, blood pressure, and affective state. Considering these potential side effects of long-term high-dose zinc and amino acid supplementation, in the current study, we proposed and validated low-dose dietary supplement mixtures for the long-term treatment of various ASD conditions.

In addition to reducing the dose of individual nutrients in supplement cocktails, our study also supports that a combination of zinc, BCAA, and serine improves synaptic function, providing broader therapeutic effects for various ASD-linked conditions relative to single-nutrient treatments. Zinc, BCAA, and serine all target to synapses and their functions are clearly interconnected (Fig 1). Therefore, a mixture containing zinc, BCAA, and serine not only exerts synergistic effects, as described herein, but is also likely to benefit other mutant mouse lines presenting synaptic impairments. In summary, nutrient cocktails that contain reduced doses of zinc, BCAA, and serine are potentially safer treatment options for long-term applications against various ASD conditions involving impaired synaptic formation and/or signaling.

## Conclusions

Through a combination of proteomic analysis, in vivo calcium imaging, and behavioral assays, we have demonstrated that supplementation with cocktails containing low doses of multiple nutrients can restore protein expression profiles and neural connectivity in the brains of multiple ASD mouse models, thereby improving their social behaviors. These findings have revealed a promising new avenue for dietary therapy in the treatment of ASD.

## Methods

### Ethics statement

All animal experiments were performed with the approval of the Academia Sinica Institutional Animal Care and Utilization Committee (Protocol No. 18-10-1234 and 23-03-1990) and in strict accordance with its guidelines and those of the Council of Agriculture Guidebook for the Care and Use of Laboratory Animals.

### Animals

The *Tbr1*[+/−], *Nf1*[+/−], and *Cttnbp2* M120I mutant mice had been generated and/or characterized previously [14,18,26,39,58,59]. Male mice were used based on previous studies [14,18,26,39,58,59]. All mouse lines were maintained by backcrossing to C57BL/6JNarl purchased from the National Laboratory Animal Center, Taiwan. Mice were group housed (<6 mice per cage) in the specific pathogen-free animal facility of the Institute of Molecular Biology, Academia Sinica, under controlled temperature (20–23 °C) and humidity (48%–55%) with free access to water and food (LabDiet

#5K54 https://www.labdiet.com/getmedia/2c493c25-079b-4b38-b2b7-6e64f1313811/5K54.pdf?ext=.pdf). One week before behavioral tests, mice were moved from the breeding room to the experimental area. Mice were further housed individually one week before conducting social behavior assays. A 12 h light/12 h dark cycle [light (intensity 240 l×): 7 a.m. to 7 p.m.] was set in the experimental area. Temperature and humidity in the test room were also controlled. All experiments were performed during the daytime, i.e., from 10 a.m. to 5 p.m.

## Cocktail supplementation

Dietary supplementation in drinking water was conducted as described previously [14,26] with some modifications. It was initiated one week before social tests or from weaning. Mice were regularly fed on Labdiet 5K54 containing Ile 0.74%, Leu 1.49%, Val 0.87%, Ser 0.83%, Zinc 81 ppm, and others (https://www.labdiet.com/getmedia/2c493c25-079b-4b38-b2b7-6e64f1313811/5K54.pdf?ext=.pdf). The average chow intake and water drinking amount are ~3 g and 4.8 ml per mouse per day, respectively. With supplementation of 1/2 cocktail (containing Ile 0.25%, Leu 0.45%, Val 0.25%, 1% serine, and 20 ppm zinc), the total intakes of these five nutrients were similar to the dietary chow containing Ile 1.1%, Leu 2.21%, Val 1.23%, Ser 2.43%, and Zinc 113 ppm. With 1/4 cocktail (containing Ile 0.1125%, Leu 0.225%, Val 0.1125%, 1% serine, and 20 ppm zinc) in drinking water, the total intakes of these five nutrients were similar to the dietary chow containing Ile 0.92%, Leu 1.85%, Val 1.05%, Ser 2.43%, and Zinc 113 ppm. Combined with 1/8 cocktail (containing Ile 0.056%, Leu 0.1125%, Val 0.056%, 0.5% serine, and 20 ppm zinc), the total intakes of these five nutrients were similar to the dietary chow containing Ile 0.83%, Leu 1.67%, Val 0.96%, Ser 1.63%, and Zinc 113 ppm. Fresh cocktail was replenished every other day throughout the assay period.

## Proteomic analysis

**LC–MS–MS.** Whole brains of mice treated with cocktail for one week were homogenized using a tissue Dounce homogenizer with a loose pestle in 1 mL sucrose buffer [50 mM Tris-Cl pH 7.4, 320 mM sucrose, 2 mM DTT, 2 mg/ml leupeptin, 2 mg/ml pepstatin-A, 2 mg/ml aproteinin, 2 mM PMSF, 2 mg/ml MG132]. The protein concentrations of total homogenate were determined by Bradford assay (Bio-Rad Protein Assay Dye Reagent Concentrate, Cat #5000006). Five mg of protein was loaded for 4% SDS-PAGE. After the dye front had completely entered the gel (about 10 min), the gel was fixed with 50% methanol and 10% acetic acid for 10 min and stained for a further 3 min with 0.1% Brilliant Blue R (B7920, Sigma-Aldrich) in 50% methanol and 10% acetic acid. The gel was then destained by destaining buffer [10% methanol, 7% acetic acid] until the bands were clearly visible. Gel bands were excised and cut into small pieces for trypsin digestion. The digestion products were subjected to LC–MS–MS using a Thermo Orbitrap Fusion Lumos mass spectrometer (Thermo-Fisher Scientific, Bremen, Germany). Proteomic data were searched against the Swiss-Prot *Mus musculus* database (17,049 entries total) using the Mascot search engine (v.2.6.2; Matrix Science, Boston, MA, USA) through Proteome Discoverer (v 2.2.0.388; Thermo-Fisher Scientific, Waltham, MA, USA), and peak-not-found groups were removed. The mass spectrometry proteomics data have been deposited to the ProteomeExchange Consortium via the PRIDE partner repository [60,61] with the dataset identifier PXD069666.

**PyWGCNA.** WGCNA is an unsupervised algorithm for finding modules of highly correlated genes/proteins [62]. The adjacency matrix was measured based on co-expression pattern similarity between the nodes (i.e., genes/proteins). After processing by sample clustering and choosing the soft-thresholding power, modules were identified as clustered interconnected nodes (genes/proteins) using hierarchical clustering. The module-trait heatmap was then identified to represent correlations of the module eigengenes with differential traits. High absolute correlation values indicated that the eigengenes have a higher probability of being affected by a trait. GO enrichment using genes identified from individual modules was further analyzed to determine the biological process pathways related to these modules.

**Principal Component Analysis (PCA) and GO analysis using STRING.** STRING analysis (https://string-db.org/) was employed to identify functional protein networks. Lines between nodes indicate the interactions based on experimental or STRING database evidence. PCA was conducted using the jupyter notebook editor with Python3.8. The

results of LC–MS–MS were loaded using the *pandas* module. The principal components were analyzed using the *sklearn* PCA module and the plot was generated using the *seaborn* module.

**Immunoblotting validation.** Sample preparation was as described above for LC–MS–MS. Protein samples (20 μg) in each well were separated by 6% or 10% SDS-PAGE and then transferred to a PVDF membrane. Membranes were rinsed with washing buffer (0.1% Tween-20 in TBS) and blocked using blocking buffer (5% nonfat milk and 0.1% Tween-20 in TBS) for 30 min and hybridized with primary antibodies in antibody dilution solution (2% horse serum, 2% bovine serum albumin, 0.1% Tween-20, 150 mM NaCl, and 10 mM Tris pH 8.0) overnight at 4 °C or for 2 h at room temperature. After washing three times, horseradish peroxidase-conjugated secondary antibodies in the antibody dilution solution were used to detect primary antibody by incubation for 1 h. After washing again three times, the results were visualized using the ECL reagents with a luminescent image analyzer. The following antibodies were used: GRIA1 (AB1504; EMD Millipore), GRIN2A (07-632; EMD Millipore), GRIN2B (AB1557P; EMD Millipore), DPYSL3 (AB5454; EMD Millipore), HOMER1 (H-174; Santa Cruz), GABBR2 (AB75838; Abcam), SYNPO (163002; SYSY), LIN7 [63], and SAP97 [64].

## Behavior analyses and cocktail supplementation

The mouse behavioral tests including an open-field test, elevated plus maze, RSI, three-chamber test, and cued fear conditioning were conducted as described previously [14,17,65,66].

**Open-field test.** The experiment was performed based on a previous study [65]. In brief, a mouse was placed into an open box (40×40×30 cm) and allowed to freely explore the box for 10 min. The area of the central region was equal to the total area of the four corners, and the regions were marked on the bottom of the box. The total moving distance and the time spent in the four corners and the central area were quantified using the Smart Video Tracking System (Panlab, Barcelona, Spain). Total travel distance indicated horizontal locomotor activity. The time spent in the corners indicated anxiety.

**Elevated plus maze.** As described previously [65], the maze consisted of two open arms (30×5 cm) and two closed arms (30×5 cm) enclosed by a 14-cm-high wall, which was used to analyze the anxiety of mice. Mice were placed individually into the central area facing one of the open arms and allowed to freely explore the maze for 10 min. Their movements were recorded by video-recording from above and analyzed using the Smart Video Tracking System (Panlab, Barcelona, Spain). The percentage of time spent in the open arms, the closed arms, and the central square was measured to evaluate the degree of anxiety of the test mice.

**Reciprocal social interaction test.** This experiment was performed as described previously [39]. An unfamiliar "stranger" wild-type male mouse that was 1–2 weeks younger than the test mouse was put into the home cage of the isolated test mouse. The unfamiliar mice had their backs shaved to distinguish them from the test mice. With the lid of the cage open, the social interaction of the test mouse with the unfamiliar mouse was recorded for 5 min using a digital camera. The time that the test mouse spent interacting with the unfamiliar mouse was then measured. Longer interaction times indicated better social behaviors.

**Three-chamber test.** This assay was conducted as described previously [39]. The entire test comprised three 10-min sessions with 5-min intervals between sessions. The entire three sessions were video-recorded from above. During the interval, test mice were placed back in their home cages. The first session was habituation when a test mouse was placed into the middle chamber to freely explore all three empty chambers. In the second session for testing sociability, stranger mouse 1 (S1) was placed in a wire cage on the opposite side of the test mouse's preferred chamber. An object (O) of similar size to the mouse was put into a wire cage on the other side. In the last session for evaluating novelty preference, stranger mouse 2 (S2) was placed in a wire cage at the opposite end of the three-chamber apparatus to S1 and the test mouse was placed into the middle chamber again to freely interact with S1 and S2. The time spent interacting or sniffing each wire cage of the left and right chambers was measured using the Smart VideoTracking System (Panlab) without knowing the genotype of the mice. The values of S1/(S1+O) and S2/(S2+S1) were determined as preference indices of sociability and social novelty preference, respectively.

**Cued fear conditioning.** Fear conditioning was performed as described previously using the FreezeScan 2.0 system (CleverSys) [39]. The entire task was carried out over 4 days. The test mouse was subjected to habitation in recording cage A for 10 min on the first 2 days. On the third day, the test mouse was placed in recording cage B for 4 min, before receiving a 20-s auditory cue with an electric shock (0.6 mA for 2 s) for the last 2 s. This cue was provided three times with a 1-min interval between each cue. On the final day, the test mouse was placed back in recording cage A for 4 min and then the freezing response was measured for a total of 20 auditory stimulations (each of 20 s) with 5-s intervals. The percentage of freezing responses to the first four stimulations was averaged to represent the degree of memory. Freezing responses were videotaped and measured using the FreezeScan 2.0 system (CleverSys).

## In vivo calcium imaging of the BLA

In vivo calcium imaging and analyses using multiple approaches were conducted as described previously [47–50,67]. Detailed information about reagents, AAV, endominiscopy, behavioral tests for in vivo calcium imaging, and all calcium imaging analyses are also summarized below.

**Reagents and AAV plasmid.** The reagents and plasmid used were as follows: 2,2,2-Tribromoethanol (Sigma-Aldrich, #T48402); Zoletil 50 (Virbac, 5 mL/vial); Xylazine hydrochloride (Sigma-Aldrich, #X1251); Sevatrim (Swiss Pharmaceutical Co. LTD., 5 mL/vial); Carprofen (Zoetis, 20 mL); Super-Bond C&B kit (Sun Medical Co., Moriyama, Shiga, Japan); pGP-AAV-syn-jGCaMP7c variant 1513-WPRE (the plasmid was a gift from Douglas Kim and the GENIE Project, Addgene plasmid #105321; http://n2t.net/addgene:105321; RRID:Addgene_105321) [68].

**Viral vector injection and GRIN lens implantation.** Three-month-old mice were anesthetized by intraperitoneal administration of 5 mL/kg of an anesthetic mixture (Zoletil, 4 mg/mL; Xylazine, 2 mg/mL). To record BLA neuronal activity, mice were injected with 0.3 μl of AAV8-Syn-jGCaMP7c variant 1513-WPRE (1.77 x $10^{13}$ vg/mL) into the right BLA (AP:1.33, ML: 3.25, DV: 4.95) at 30 nl/min using the Hamilton syringe system. Two weeks later, a second surgical procedure was performed to implant the gradient index (GRIN) lens. The tissue along the path above the injection site was gently removed using a 21G sterile needle. The 0.5-mm-diameter GRIN lens (efocus Imaging Cannula Model L, Doric Lenses, Québec, Canada) was implanted in a position above the jGCaMP7c injection site (AP:1.33, ML: 3.25, DV: 4.9) to observe neuronal activity. The GRIN lens was then fixed using Super-Bond (Sun Medical Co.). Antibiotics (Sevatrim, 15 TMP/kg) and an analgesic drug (Carprofen, 5 mg/kg) were injected intravenously into mice for the following 7 days. The mice underwent four weeks of recovery following surgery before undergoing behavioral testing and recording.

**Behavioral tests for in vivo calcium imaging.**

*Open field test (OF)*—Mice hosting a GRIN lens and miniscope (eSFMB_L_458, Doric Lenses, Québec, Canada) were individually placed in a transparent plastic box (40 cm (w) × 40 cm (d) × 30 cm (h)) for free exploration for 10 min. The total area of the four corners was equal to that of the center. Mouse behaviors were recorded from above using a camera (BTC_USB3.0, Doric Lenses, Québec, Canada). The behavior videos were analyzed suing the Smart Video Tracking System (Panlab, Barcelona, Spain). The total distance of movement and the ratio of time spent in the corners to that in the center were calculated to evaluate the locomotion and anxiety of mice.

*Object exploration (OE)*—A toy (of similar size to a mouse) was put into the center of the home cage of an isolated tested mouse. A mouse hosting a GRIN lens and miniscope was free to explore the toy for 10 min. OE includes approaching, sniffing, and touching. The total interaction time and the time points of starting and ending OE were recorded for further analysis with in vivo calcium imaging results.

*Reciprocal social interaction (RSI)*—A younger unfamiliar mouse (2-month-old) was placed into the home cage of a test mouse for 10 min. The interaction between the unfamiliar mouse and the test mouse hosting a GRIN lens and miniscope was recorded from above. Approaching, sniffing, social grooming, mounting, and chasing initiated by the test mouse were all coded as social interactions. The total interaction time and the time points of starting and ending the social interactions were recorded for further analysis with in vivo calcium imaging results. Only behaviors initiated by the test mouse were

counted. Aggressive behaviors (attack) and passive social interactions (unfamiliar mouse actively touching the experimental mouse) were not included in the analysis.

**Recording of neuronal activity and signal extraction.** Calcium imaging (i.e., neuronal activity) and mouse behaviors were synchronously recorded at a frame rate of 10 Hz using the efocus Fluorescence Microscope System (Doric Lenses , Québec, Canada). Calcium transients were acquired in a field of view of 630 × 630 pixels (320 × 320 µm) with current 30–100 mA (76–200 µW at the objective, 465 nm) and analog gain of 1. The calcium image was then spatially downsampled by a factor of nine and cropped into 180 × 180 pixels. CaImAn (1.8.5, https://github.com/flatironinstitute/CaImAn) [67], an open-software package, was used for motion correction and signal extraction. The non-corrected regions of interest were removed manually. The activity of each neuron was outputted and then normalized via z-score transformation.

**Confirmation of injection site and GRIN lens position.** After completing the entire recording, test mice were deeply anesthetized via intraperitoneal injection of 0.7–0.8 mL 2,2,2-Tribromoethanol per mouse, then transcardially perfused with phosphate-buffered saline (PBS), followed by 4% paraformaldehyde in PBS. The brains were collected and post-fixed overnight at 4 °C. After two days of dehydration in 30% sucrose in PBS, the brains were cut into slices of 60 µm thickness by cryosectioning and stored in PBS. Slices were washed in TBS (25 mM Tris-Cl pH 7.5, 0.85% NaCl) three times (10 min each) and stained with DAPI (1 µg/ml). After mounting, slices were imaged using a microscope (AxioImager-Z1; Carl Zeiss).

**Correlation analysis between behaviors and neuronal ensembles.** The relevance of behaviors to neuronal ensembles was analyzed as described previously [47]. In brief, the within-cluster sum of squares (WCSS) was used to evaluate the potential number of neuronal clusters. Neurons were then separated into different clusters via *k*-means clustering. Pearson correlation was used to evaluate two types of correlations. The first was the correlation between the behavior vector (binary vector; social versus non-social period or object-approaching versus non-object-approaching period) and mean neuronal ensemble activity. The second was the correlation of the activities between neurons in the same ensemble. To determine if the neurons retained ensemble membership across different sessions, the correlation between the activity of an individual neuron and mean activity of the ensemble to which it belonged was calculated [47]. The multi-session registration function in CaImAn was used to identify the same neurons across sessions [67].

**Correlation analysis between behaviors and individual neurons.** To define behavior-relevant cells at the single-cell level, the cosine similarity between the behavior vector (binary vector; social versus non-social period or object-approaching versus non-object-approaching period) and neuronal activity greater than the 95th percentile or lower than the 5th percentile of randomly shuffled data (1,000 permutations) was recognized as neurons positively or negatively related to specific behavior, respectively [48]. The cosine similarity of neurons lying between the 95th and 5th percentiles of shuffled data defined neurons irrelevant to social behavior [48]. The cosine similarity ranges from 0 to 1.

The ratio of social behavior-relevant cells in WT and *Tbr1*$^{+/-}$ mice across sessions was calculated to examine changes in population. The number was normalized against total cell number (after multi-session registration). To investigate the response of social behavior-relevant neurons during interactions, the mean activity of OE-related positively correlated neurons when approaching the object and sociality-related positively correlated cells during social interactions was calculated. To track cell identity across sessions, the same neuron was identified via multi-session registration using CaImAn [67] and connectograms (Circos) were used to represent changes in identity.

**Functional network analysis.** To evaluate neuronal activation patterns among different neurons, cosine similarity was used to calculate the correlation of neuronal activities between neurons. Phase randomization was used to randomize (10,000 times) the original neuronal activity of each pair [50]. The cosine similarity between two neurons greater than the 99.17th percentile of randomly shuffled data was recognized as highly correlated cells (significant similarity) [49]. To quantify the network structures, the degree centrality of a node (neuron) was used to calculate the total number of links with other nodes using the NetworkX Python package [49]. Links represent significant similarity between two neurons.

## Statistical analysis

All quantitative data in this report are presented as means ± SEM. The individual data points are also shown. Graphs were plotted using GraphPad Prism 7.0 or 10.0 (GraphPad software). Although no randomization was performed to allocate subjects in the study, mice were arbitrarily assigned to different treatments. No statistical method was applied to evaluate the sample size, but our sample sizes are similar to previous publications [14,42]. To avoid potential personal bias in behavioral analyses, the data were collected and analyzed blindly without knowing the genotype or treatment of the mice. For behavior assays, statistical analysis was performed using a two-tailed unpaired Student $t$ test for two-group comparisons, and two-way ANOVA and Bonferroni's correction was performed for two-factor four-group comparisons. A Kolmogorov–Smirnov test in the Scipy Python package was used to determine cumulative probabilities. $P$ values of less than 0.05 were considered significant. * $P < 0.05$; ** $P < 0.01$; *** $P < 0.001$; # $P < 0.05$; ## $P < 0.01$; ### $P < 0.001$. All numerical values underlying the figures are available in the Source Data (S3 Data), and all statistical methods and results are summarized in S4 Data.

## Supporting information

**S1 Fig. Pre-processing workflow of the Python package for weighted correlation network analysis (PyWGCNA).** (**A**) Sample clustering. The three samples of the Tbr1_Water group were closer to each other and distinct from the other samples. (**B**–**C**) Checks of the soft-thresholding power for network topology analysis. (**D**) Clustering of module eigengenes. The cutoff value of distance was set at 0.2. Thus, modules with a distance < 0.2 were merged into one. Related to Fig 1. The entire list of the selected proteins for PyWGCNA can be found in S1 Data.
(TIF)

**S2 Fig. Gene ontology of the PyWGCNA results.** (**A**) Black, (**B**) Dark gray, (**C**) Indian red, and (**D**) Light gray eigengenes. Related to Fig 1. All selected proteins of Black module are available in S2 Data.
(TIF)

**S3 Fig. Gene ontology of the PyWGCNA results (continued).** (**A**) Brown, (**B**) Gainsboro, and (**C**) Whitesmoke eigengenes. Related to Fig 1.
(TIF)

**S4 Fig. Protein subnetwork of the Black module.** (**A**) All synapse-related GO (70 proteins). (**B**) Cytoskeleton organization-related GO (43 proteins). (**C**) Proteins related to the GO term of generation of precursor metabolites and energy (24 proteins). (**D**) Vesicle-mediated transport GO (47 proteins). Related to Fig 1.
(TIF)

**S5 Fig. Confirmation of GCaMP7c expression and GRIN lens position, as well as registration of the same neurons across sessions.** (**A**) Schematic of virus injection and GRIN lens implantation. (**B**) An example of a mouse brain infected with AAV and implanted with a GRIN lens. (**C**) Positioning of the GRIN lens in WT and *Tbr1*$^{+/-}$ mice (WT: $n = 4$; *Tbr1*$^{+/-}$: $n = 4$). D, dorsal; V, ventral; M, medial; L, lateral. (**D**) Number of firing neurons in WT and *Tbr1*$^{+/-}$ mice across sessions. Data is represented as mean ± SEM. Each dot represents an individual mouse. (**E**–**G**) Processing of calcium images using CaImAn. (**E**) Template for motion correction. (**F**) Accepted firing neurons are labeled by white circles. (**G**) The same accepted firing neurons after multi-session registration are labeled by red circles. Related to Fig 4. The data underlying the graph of (**D**) can be found in S3 Data. The statistical results are available in S4 Data. The figure was *created in BioRender. Lin, M. (2025) https://BioRender.com/akrum83*.
(TIF)

**S6 Fig. Two neuronal ensembles of BLA firing neurons encode object exploration.** (**A**) The firing neurons during object exploration (OE) in the WT_Water, *Tbr1*$^{+/-}$_Water, WT_Cocktail, and *Tbr1*$^{+/-}$_Cocktail groups were separated into

two neuronal ensembles via within-cluster sum of squares (WCSS) analysis. Data is represented as mean ± SEM. (**B**) Clustermap was used to represent Pearson's correlations among neuronal activity during object exploration (OE). Related to Fig 4.
(TIF)

**S7 Fig. Two neuronal ensembles of BLA firing neurons encode reciprocal social interaction (RSI).** (**A**) The firing neurons during RSI in the WT_Water, *Tbr1*$^{+/-}$_Water, WT_Cocktail, and *Tbr1*$^{+/-}$_Cocktail groups were separated into two neuronal ensembles via within-cluster sum of squares (WCSS) analysis. Data is represented as mean ± SEM. (**B**) Cluster map was used to represent Pearson's correlations among neuronal activity during RSI. Related to Fig 4.
(TIF)

**S8 Fig. Network connectivity of BLA firing neurons during reciprocal social interaction (RSI).** Functional networks of BLA firing neurons of individual mice (WT: $n = 4$; *Tbr1*$^{+/-}$: $n = 4$) in RSI. Connections (lines) between the nodes indicate a significant similarity in activation patterns relative to shuffled data (10,000 permutations via phase randomization). Red, neuron positively correlated with sociality. Blue, neuron negatively correlated with sociality. Gray, neuron irrelevant to social behavior. WT mouse #3 and *Tbr1*$^{+/-}$ mouse #4 are also shown in Fig 6B. The data underlying the graphs shown in this figure can be found in the S3 Data.
(TIF)

**S9 Fig. Network connectivity of BLA firing neurons during object exploration.** Functional networks of BLA neurons identified during the optical exploration (OE) test. Connections (lines) between nodes indicate a significant similarity in activation patterns relative to shuffled data (10,000 permutations via phase randomization). The red and blue nodes represent neurons positively or negatively associated with object exploration. The gray nodes represent neurons irrelevant to approaching behaviors. WT mouse #3 and *Tbr1*$^{+/-}$ mouse #4 are also shown in Fig 6D. The data underlying the figure can be found in the S3 Data.
(TIF)

**S1 Data. The results of LC–MS–MS. The first sheet contains the IDs and normalized protein levels of all identified proteins.** The second sheet provides sample grouping information, and the third sheet lists all proteins selected for PyWGCNA analysis. Related to Fig 2. The raw data of mass spectrometry are available via the ProteomeXchange Consortium with identifier PXD069666.
(XLSX)

**S2 Data. The results of the Black module of PyWGCNA.** The connectivity, as well as the mean and individual relative levels of selected proteins within the Black module, are listed. Related to Fig 2.
(XLSX)

**S3 Data. Source data.** Related to Figs 3B, 4B–4F, 5B, 5D, 6B–6E, 7B, 7D, 8B–8G, 8I, S5D, S8, S9.
(XLSX)

**S4 Data. Statistical results.** Related to Figs 3–8.
(XLSX)

**S1 Raw Images. Full scans of immunoblots corresponding to Fig 3A.**
(PDF)

## Acknowledgments

We thank Academia Sinica Common Mass Spectrometry Facilities for Proteomics and Protein Modification Analysis located at the Institute of Biological Chemistry, Academia Sinica (supported by AS-CFII-108-107), the Mass Spectrometry

Facility of the Genomics Core at the Institute of Molecular Biology, Dr. John O'Brien for English editing, and members of Y.-P.H.'s laboratory who relabeled samples for blind experiments.

## Author contributions

**Conceptualization:** Tzyy-Nan Huang, Ming-Hui Lin, Yi-Ping Hsueh.

**Funding acquisition:** Yi-Ping Hsueh.

**Investigation:** Tzyy-Nan Huang, Ming-Hui Lin, Tsan-Ting Hsu.

**Methodology:** Tzyy-Nan Huang, Ming-Hui Lin, Tsan-Ting Hsu, Chen-Hsin Yu.

**Project administration:** Yi-Ping Hsueh.

**Supervision:** Yi-Ping Hsueh.

**Visualization:** Tzyy-Nan Huang, Ming-Hui Lin.

**Writing – original draft:** Tzyy-Nan Huang, Ming-Hui Lin, Yi-Ping Hsueh.

**Writing – review & editing:** Tzyy-Nan Huang, Ming-Hui Lin, Tsan-Ting Hsu, Chen-Hsin Yu, Yi-Ping Hsueh.

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
