## [Editor Report · Decision Letter 0]

27 May 2025

Dear Yi-Ping,

Thank you for submitting your manuscript entitled "Abnormal synaptic proteomes, impaired neural ensembles, and defective behaviors in autism mouse models are ameliorated by dietary intervention with nutrient mixtures" for consideration as a Research Article by PLOS Biology.

Your manuscript has now been evaluated by the PLOS Biology editorial staff as well as by an academic editor with relevant expertise and I am writing to let you know that we would like to send your submission out for external peer review.

Once your full submission is complete, your paper will undergo a series of checks in preparation for peer review. After your manuscript has passed the checks it will be sent out for review. To provide the metadata for your submission, please Login to Editorial Manager (https://www.editorialmanager.com/pbiology) within two working days, i.e. by May 29 2025 11:59PM.

Kind regards,

Luke

Lucas Smith, Ph.D.

Senior Editor

PLOS Biology

lsmith@plos.org

---

## [Decision Letter · Decision Letter 1]

31 Jul 2025

Dear Yi-Ping,

Thank you again for your patience while your manuscript "Abnormal synaptic proteomes, impaired neural ensembles, and defective behaviors in autism mouse models are ameliorated by dietary intervention with nutrient mixtures" was peer-reviewed at PLOS Biology. It has now been evaluated by the PLOS Biology editors, an Academic Editor with relevant expertise, and by several independent reviewers.

In light of the reviews, which you will find at the end of this email, we would like to invite you to revise the work to thoroughly address the reviewers' reports.

As you will see below, the reviewers find the study to be generally well done, however they have each raised a number of important points including textual, methodological, and analytical concerns - and we think these should be thoroughly addressed in a revision before we can consider your study for publication. I note that reviewer 2 has additionally questioned whether the findings are sufficiently novel for PLOS Biology. While we appreciate this reviewer's concerns, and think the novelty of the findings could be made more explicit in the discussion, having discussed this with the other reviewers in our cross reviewer commenting and with the Academic Editor, we do not share this concern. We think the findings do go sufficiently beyond previous work in this area, and so we think the study is a reasonable candidate for publication at PLOS Biology assuming the other concerns are adequately addressed. As mentioned though, to address this point we would suggest that you strengthen the discussion by adding additional explicit comparisons of the effects of dietary supplement cocktail to single nutrient treatments (perhaps answering the questions laid out in reviewer 2's point 11)

Given the extent of revision needed, we cannot make a decision about publication until we have seen the revised manuscript and your response to the reviewers' comments. Your revised manuscript is likely to be sent for further evaluation by all or a subset of the reviewers.

**IMPORTANT - SUBMITTING YOUR REVISION**

*Re-submission Checklist*

*Published Peer Review*

*PLOS Data Policy*

*Blot and Gel Data Policy*

Sincerely,

Luke

Lucas Smith, Ph.D.

Senior Editor

PLOS Biology

lsmith@plos.org

REVIEWS:

Reviewer #1: This is a very strong study that shows the role of nutrients in restoring ASD-linked deficits in the Tbr1+/- mouse model. Huang et al. provides a compelling narrative on the benefits of nutrient supplements with BCAA, L-Serine and Zinc for improving social behaviors in the Tbr1+/- mouse model. Through a combination of proteomics, calcium imaging, and behavioral imaging, Huang et al. highlight the behavioral improvements seen in Tbr1+/- mice upon administration of this combined nutrient supplement.

Major results from this set of experiments are as follows:

1. Proteomic analysis from LC-MS-MS experiments revealed a difference in the Tbr1+/- group treated only with water (as evidenced by PCA analysis in Figure 1), suggesting that cocktail administration rescues the protein composition of brain lysates (becoming more similar to the wild-type groups) in the Tbr1+/- cocktail group. Additionally, gene analysis showed that cocktail treatment affects genes involved with synapses and mitochondrial activity.

2. Calcium imaging revealed that: (1) Tbr1+/- mice have stronger correlations between individual neurons and their ensembles, regardless of the treatment or task (in both open exploration and reciprocal social interaction), (2) Tbr1+/- mice treated with the nutrient supplement had a decrease in the positively-correlated neurons and an increase in the negatively correlated neurons in the reciprocal social interaction task, (3) Tbr1+/- mice treated with the nutrient supplement mixture have decreased mean activity in reciprocal social interaction, suggesting that this mixture reduces hyperactivity, and (4) the neurons that respond to the reciprocal social interaction task change as a result of nutrient supplementation. From these results and the additional information shared in the paper, the authors have clear evidence that points towards a rescue of the hyperactivity seen in the BLA network as a result of supplementation with this mixture of nutrients in Tbr1+/- mice.

3. Lastly, using behavior experiments, the authors confirm that the mixture of nutrients at lower doses has superior effects to using individual nutrients in the reciprocal social interaction task. They demonstrate this finding in other mouse models as well, including the Cttnbp2+/M120I model. They also perform other behavioral experiments - including the three-chamber test, the auditory fear conditioning test, as well as the open field and elevated plus maze tests. Their findings are in line with the references they cite, suggesting that there is an added benefit to combining these nutrients instead of providing supplementation with individual nutrients.

Overall, this study includes controls to show that the effects seen in the Tbr1+/-mice are valid. Although the authors state that there were no calculations done to determine sample sizes, it seems as if there are sufficient sample sizes for each experiment, and the authors state that they have used similar sample sizes in previous studies.

Minor weaknesses:

1. In the introduction, an additional explanation for the selection of the two additional ASD mouse models would be helpful to understand the rationale behind selecting those specific models as positive controls.

2. The authors should revise this paper to ensure that figures are referenced within the correct sections. For example, in the paragraph 233-239, the reference to Figure 3E should be to Figure 2E. Similarly, in the following paragraph (Lines 240-250), both references to Figure 3F should be to Figure 2F. Again, in the next result section (lines 258-270), the reference to Figure 4C should be to Figure 3C.

Other than these minor comments, this paper provides a strong narrative for the benefit of combined nutrient supplement in correcting the social behaviors seen in the Tbr1+/- ASD mouse model.

Reviewer #2: Based on numerous prior studies from this group and others showing the beneficial effects of single supplements (serine, BCAA, zinc) in ASD models (ref. 10, 12-18, 23-27), the current study tested the effects of a cocktail of dietary supplements using proteomics, in vivo Ca2+ imaging and behavioral assays. While the experiments are carefully designed and conducted, the conceptual novelty of this study is limited. Furthermore, there are a few concerns that need to be addressed.

1. Proteomics studies focus on one of the co-expression modules (Black), and did GO and STRING analyses (Fig. 1). However, it is unclear how these synaptic proteins are altered by Tbr1 haploinsufficiency and cocktail supplementation. Are they significantly reduced in Tbr1+/- mice and rescued by cocktail supplementation? The expressional changes in key proteins need to be confirmed.

2. The conclusion that "the total proteomes of Tbr1+/- mouse brains are different from those of WT brains and that cocktail supplementation makes the protein expression profiles of Tbr1+/- mice more comparable to those of WT mice" is mainly based on PCA plots, which is not adequate. More in-depth analyses are needed to show the impact of cocktail supplementation on protein expression in this ASD model.

3. Fig. 2, what is the relevance of the "stronger correlation between individual BLA neuron activity and its corresponding ensemble" in Tbr1+/- mice to their behavioral deficits?

4. Fig. 3C, it is unclear whether the differential neuronal responses to social stimulation by Tbr1 haploinsufficiency or cocktail supplementation are statistically and biologically significant. How could Tbr1 haploinsufficiency alter the cell type responding to social stimulation (line 458-459)?

5. Fig. 3E and 3F, the extremely low number of responsive BLA neurons shared by RSI-1 and RSI-2 in WT (7 neurons) and Tbr1+/- (1 neuron) raise concerns on the data validity and interpretations.

6. How to link Tbr1 deficiency-induced "hyperactivity and hyperconnectivity of BLA neurons during social interaction" to the social deficits in Tbr1+/- mice? The degree of whole-brain synchronization (connectivity) of the BLA is reduced by Tbr1 haploinsufficiency [43]. What is the effect of dietary cocktail on the correlated neuronal activity between BLA and other brain regions?

7. Proteomic and in vivo Ca2+ imaging tests were only done in one ASD model (Tbr1+/- mice). The title and abstract gave the impression that dietary cocktail ameliorated the altered proteins and neural connectivity across multiple ASD models, which needs to be corrected. It is simply unknown whether other ASD models have abnormal synaptic proteomes and impaired neural ensembles in BLA that are similar to Tbr1+/- mice, and whether the dietary mixtures have any effects on proteomes and neural ensembles in these models.

8. The authors demonstrated previously that 0.45% BCAA or 40 ppm zinc supplemented is sufficient to improve the social behaviors of Cttnbp2+/M120I mice [13, 14, 18], while this study showed the lack of effects with .225% BCAA or 20 ppm zinc (Fig. 5D). What makes such a big difference by just reducing the dose by half?

9. Fig. 6E, is there any significant difference between Tbr1+/- with water and Tbr1+/- with ¼ cocktail? If not, the conclusion "Thus, the 1/4 cocktail treatment rescued the memory performance of Tbr1+/- mice" is overrated.

10. Why were 1/4 cocktail, 1/8 cocktail or 1/2 cocktail used in behavioral assays (Fig. 5, & 6)? Do different ASD models need different doses of the supplement mixture? If so, what is the potential reason?

11. Given what has been extensively reported on single supplements, the current study does not provide new insights into the significant benefits of dietary mixtures. What are the side effects of single supplements? What makes dietary mixtures more beneficial? What is the reason underlying the better effects of dietary mixtures than single supplements?

12. There are a few errors when referring to Figures (e.g. should be Fig. 2E (line 238), Fig. 2F (line 246), Fig. 2F (line 248), Fig. 3C (line 268)).

Reviewer #3: This is an interesting and well-executed manuscript demonstrating that Tbr1 haploinsufficiency leads to hyperactivation and increased neuronal connectivity within the basolateral amygdala (BLA). To counteract these abnormalities, the authors developed nutritional supplement cocktails containing zinc, branched-chain amino acids (BCAAs), and serine—nutrients known to influence synapse formation and neuronal function—which were administered to Tbr1+/- mice. This intervention altered both the synaptic and metabolic proteomes and restored BLA activity and functional connectivity during social interactions. Importantly, while low doses of individual nutrients had no significant effect on social behaviour, their combination in supplement form significantly improved social behaviour and associative memory in Tbr1+/- mice, suggesting a synergistic effect. Moreover, the supplement cocktails also ameliorated social deficits in two additional ASD mouse models, Nf1+/- and Cttnbp2+/M120I, indicating potential for broader therapeutic application.

While the experimental design is generally sound, a critical issue should be addressed before the manuscript is suitable for publication. Specifically, although the authors compellingly demonstrate BLA hyperactivity in the Tbr1+/- model and show that the supplement cocktail modulates both neuronal activity and the synaptic proteome, the mechanistic link between these proteomic changes and the observed restoration of BLA connectivity remains unclear. Providing further clarification or additional data to establish a causal relationship between the proteomic alterations induced by the supplement and the normalisation of network-level connectivity would significantly strengthen the manuscript.

---

## [Editor Report · Decision Letter 2]

17 Oct 2025

Dear Yi-Ping,

Thank you for your patience while we considered your revised manuscript "Dietary nutrient mixtures ameliorate synaptic proteome abnormalities, neural ensemble dysfunction, and behavioral deficits in the Tbr1 mouse model of autism" for publication as a Research Article at PLOS Biology. This revised version of your manuscript has been evaluated by the PLOS Biology editors and the Academic Editor, who is fully satisfied by the changes made in this revision.

Based on our Academic Editor's assessment of your revision, we are likely to accept this manuscript for publication. However, before we can editorially accept your study, we need you to address a number of data and other policy-related requests, in a last revision that we anticipate will not take very long. These requests are detailed below.

**IMPORTANT: Please address the following editorial requests.

1) TITLE: We would like to suggest a change to the title, that we think will make the manuscript a bit more broadly appealing to our readership. If you agree, we suggest you change the title to:

"Low-dose mixtures of dietary nutrients ameliorate behavioral deficits in multiple mouse models of autism"

^We are happy to discuss the title further, if you wish.

2) COMPETING INTERESTS: Please update the 'competing interests' statement in our editorial manager system to indicate that Yi-Ping Hsueh is a member of the PLOS Biology editorial board, but that this has not impacted the review process or editorial assessment.

3) DATA: Please upload your raw mass spec data to a relevant data repository (such as ProteomeXchange's PRIDE)

4) DATA: You may be aware of the PLOS Data Policy, which requires that all data be made available without restriction: http://journals.plos.org/plosbiology/s/data-availability. For more information, please also see this editorial: http://dx.doi.org/10.1371/journal.pbio.1001797

Note that for the most part we do not require all raw data (except for where this is the standard in the field, see request above). Rather, we ask that all individual quantitative observations that underlie the data summarized in the figures and results of your paper be made available in one of the following forms:

a. Supplementary files (e.g., excel). Please ensure that all data files are uploaded as 'Supporting Information' and are invariably referred to (in the manuscript, figure legends, and the Description field when uploading your files) using the following format verbatim: S1 Data, S2 Data, etc. Multiple panels of a single or even several figures can be included as multiple sheets in one excel file that is saved using exactly the following convention: S1_Data.xlsx (using an underscore).

b. Deposition in a publicly available repository. Please also provide the accession code or a reviewer link so that we may view your data before publication.

>>Regardless of the method selected, please ensure that you provide the individual numerical values that underlie the summary data displayed in the following figure panels as they are essential for readers to assess your analysis and to reproduce it:

Fig 3B; Fig 4B-F; FIg 5B,D, Fig 6B-E; FIg 7 B,D; Fig 8B-G,I;

Fig S5D; Fig S8; Fig S9

>>Please also ensure that figure legends in your manuscript include information on where the underlying data can be found, and ensure your supplemental data file/s has a legend.

>>Please ensure that your Data Statement in the submission system accurately describes where your data can be found.

5) WESTERN BLOTS: Thank you for providing a supplemental file with the western blot images related to your paper. These images look slightly cropped, and so I ask that you update this file to include the fully uncropped images related to your study. Ideally the image should include space around the edges of the manuscript - or if it is difficult to see that, you can just provide the full page image taken by your scanner.

6) CODE: Thank you for providing a github link that includes the code related to your paper. I tried this link and for some reason it didn't work. Can you make sure it is correct?

In parallel, please note that we cannot accept sole deposition of code in GitHub, as this could be changed after publication. However, you can archive this version of your publicly available GitHub code to Zenodo. Once you do this, it will generate a DOI number, which you will need to provide in the Data Accessibility Statement (you are welcome to also provide the GitHub access information). See the process for doing this here: https://docs.github.com/en/repositories/archiving-a-github-repository/referencing-and-citing-content

We expect to receive your revised manuscript within two weeks.

*Published Peer Review History*

*Press*

Sincerely,

Luke

Lucas Smith, Ph.D.

Senior Editor

lsmith@plos.org

PLOS Biology

---

## [Editor Report · Decision Letter 3]

27 Oct 2025

Dear Yi-Ping,

Thank you for the submission of your revised Research Article "Low-dose mixtures of dietary nutrients ameliorate behavioral deficits in multiple mouse models of autism" for publication in PLOS Biology and thank you for addressing our last editorial requests in this revision. On behalf of my colleagues and the Academic Editor, Sebastien G Bouret, I am pleased to say that we can in principle accept your manuscript for publication, provided you address any remaining formatting and reporting issues. These will be detailed in an email you should receive within 2-3 business days from our colleagues in the journal operations team; no action is required from you until then. Please note that we will not be able to formally accept your manuscript and schedule it for publication until you have completed any requested changes.

PRESS

Sincerely, 

Luke

Lucas Smith, Ph.D.

Senior Editor

PLOS Biology

lsmith@plos.org